# Topographically-controlled contribution of avalanches to glacier mass balance in the 21st century

Marin Kneib [1,2,3,4] ✉, Fabien Maussion [4,5], Fanny Brun [3], Guillem Carcanade[3], Daniel Farinotti [1,2], Matthias Huss [1,2,6], Marit van Tiel[1,2], Achille Jouberton[7], Patrick Schmitt [4], Lilian Schuster [4], Amaury Dehecq [3] & Nicolas Champollion [3]

Glaciers are often located in steep mountain settings and avalanches from surrounding slopes can strongly influence snow accumulation patterns on their surface. This effect has however never been quantified for more than a few glaciers and the impact on the future evolution of glaciers is unclear. We coupled an avalanche and a glacier model to estimate the contribution of avalanches to the accumulation of all glaciers in the world and how this affects their evolution throughout the 21st century. Globally, 3% of the snow accumulation on glaciers comes from avalanches and 1% is removed by avalanches. This net contribution varies between regions and glaciers, with a maximum of 15% for New Zealand. Accounting for avalanches modifies the altitudinal pattern of glacier mass balance and the projected evolution of individual glaciers. The main effects include (1) a longer persistence of small glaciers, with for example three times more ice retained by glaciers smaller than 1 km$^2$ in Central Europe under a low-emission scenario, and (2) an increased sensitivity of high-elevation accumulation zones to future warming. We anticipate the relative influence of avalanches to increase in the future and advocate for a better monitoring of this process and representation in glacier models.

Glaciers are important components of the world's 'water towers'[1] and are currently undergoing major changes due to atmospheric warming[2,3]. Future projections show that they will lose considerable mass regionally in the coming decades[4–6], thereby increasing the vulnerability of downstream regions to future atmospheric warming[7]. However, these projections remain uncertain for individual glaciers[4] due to a lack of understanding and quantification of key processes influencing glacier mass balance, such as supraglacial debris, high-elevation precipitation and avalanching[8,9].

Mountain glaciers usually gain mass via solid precipitation falling in their accumulation area that is then advected downstream with glacier flow. Thus, the mass balance is traditionally expected to increase with elevation[10]. For catchments in steep mountain ranges however, there can be large mass inputs from mountain headwalls at localized portions of the glacier, both in the accumulation and ablation zones, which leads to a non-linear elevation distribution of mass balance[11–14]. These inputs, which vary in size and originate from the gravitational redistribution of snow or ice, can contribute to the

[1]Laboratory of Hydraulics, Hydrology and Glaciology (VAW), ETH Zurich, Zurich, Switzerland. [2]Swiss Federal Institute for Forest, Snow and Landscape Research (WSL), bâtiment ALPOLE, Sion, Switzerland. [3]Institut des Géosciences de l'Environnement, Université Grenoble Alpes, CNRS, IRD, Grenoble, France. [4]Department of Atmospheric and Cryospheric Sciences, University of Innsbruck, Innsbruck, Austria. [5]School of Geographical Sciences, University of Bristol, Bristol, UK. [6]Department of Geosciences, University of Fribourg, Fribourg, Switzerland. [7]Institute of Science and Technology Austria, ISTA, Klosterneuburg, Austria. ✉e-mail: kneibm@ethz.ch

persistence of glaciers at low altitudes[15–17] and therefore buffer the depletion of mountain water resource[18]. In the few locations where on-glacier avalanching has been estimated, it is one of the main sources of mass inputs, with estimated avalanche contributions ranging between 9% and 92% of the annual accumulation[13,14,19–22]. However, this present and future influence has never been quantified for more than a few glaciers, with large uncertainties, due to a lack of field measurements of mass contribution from avalanching and tailored remote sensing approaches[23].

Global and regional-scale models usually express the glacier mass balance as a function of temperature and precipitation[24–26], with an empirical correction factor that scales precipitation inputs and attempts to account for unobserved processes such as avalanching, snow redistribution by wind and orographic precipitation[27]. This precipitation correction factor, together with a temperature bias parameter and the melt factor used in global-scale temperature-index models, needs to be calibrated on a glacier-by-glacier basis[28]. In recent years, there has been an emergence of high-resolution large-scale geodetic estimates characterizing the state of the world's glaciers[2,3], paving the way for a much more accurate calibration of mass balance parameters for each individual glacier in regional to global-scale models. However, even with better calibration data, the non-linearity of the mass balance patterns from avalanches (among other processes; Fig. 1) is impossible to match by these simple models, and leads to an inaccurate representation of the spatial variability of mass balance and, as a result, glacier flow[10]. In such cases the precipitation correction factor can end up being particularly and unrealistically high to account for the excess mass input[29]. A more explicit representation of these nonlinear mass balance contributions from avalanching is likely to help constrain the glacier-wide mass balance parameters and therefore improve the simulations from these glacier models, as recently shown for other processes, notably calving[30] or sub-debris melt[6,8].

In this study we couple a global glacier model with a gravitational snow redistribution model to estimate the current contribution of avalanches to glacier accumulation for all glaciers in the world. We use this modelling framework to derive the influence of avalanches on the future evolution of individual glaciers and mountain regions throughout the 21st century and analyse this influence relative to model runs that do not explicitly account for avalanches. Finally, we assess how this avalanche contribution will evolve as glaciers retreat in a warming climate.

## Results and discussion

### Substantial contribution of avalanches to glacier accumulation

We use a simple gravitational snow redistribution model[31,32] run with W5E5v2.0 mean monthly climate inputs[33] over the glacier model calibration period 01/2000-12/2019 to quantify the contribution of avalanches to the annual glacier accumulation (Methods). The snow redistribution model is run at a monthly time scale and removes the snow from areas where the snowpack exceeds a certain threshold determined by the slope. The maximum allowable snow depth is set to decrease exponentially as the slope increases[31]. This implies that our definition of avalanches encompasses any type of gravitational snow redistribution from steep terrain onto lower angle slopes, which includes all magnitudes of events, from drifting snow along steep headwalls to large slab avalanches[21,34,35]. We note that this model does not represent wind-driven snow redistribution, which is a particularly difficult process to constrain given its dependence on an accurate representation of wind fields on a glacier, and despite recent advances in quantifying and modelling this effect at relatively small spatial scales[32,36–38].

The glacier-wide avalanche contribution to accumulation can vary considerably, with values above 50%, indicating that more than half of the annual snow accumulation comes from avalanches, and values lower than −50%, indicating that more than half of the snow accumulated on the glacier surface has been removed by avalanches (Fig. 2; Supplementary Fig. 1; Supplementary Table 1). For the glacierized regions of the world described in the Randolph Glacier Inventory v6.0[39], the regional avalanche contribution is generally positive, with higher values for regions prone to steeper relief and higher precipitation. New Zealand is the region with the highest net avalanche contribution, with a region-wide contribution of 14.6% or

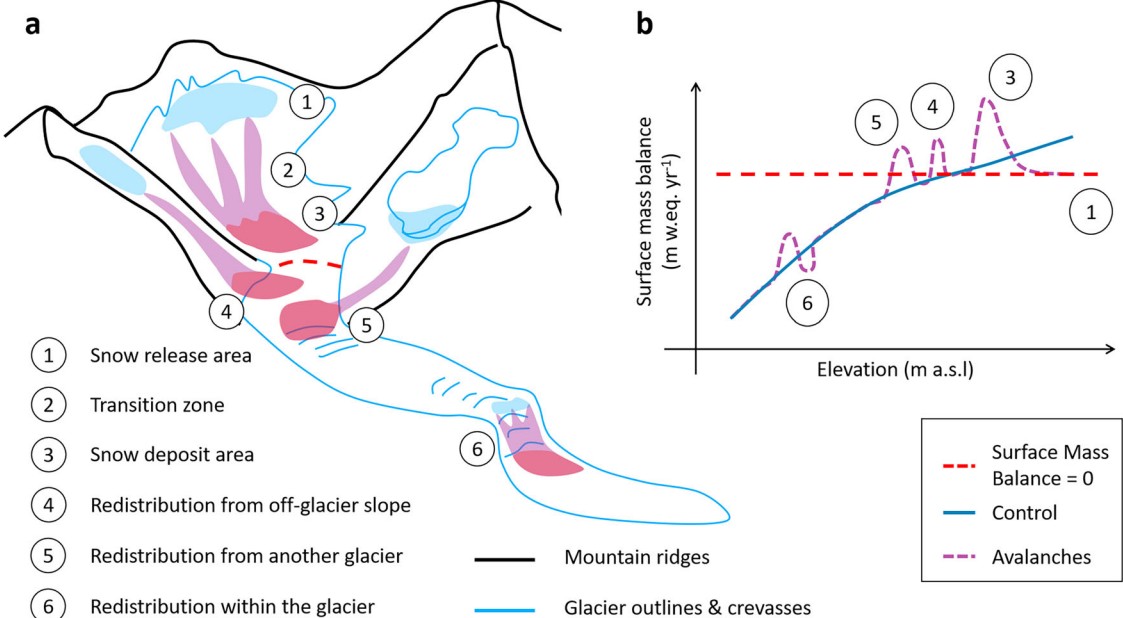

**Fig. 1 | Impact of avalanches occurring at the surface of a hypothetical mountain glacier. a** Hypothetical glacier with avalanche release, transition and deposit areas in light blue, purple and red, respectively. **b** Corresponding effect on the annual mass balance patterns with elevation. Avalanches occurring within the glacier boundaries (1-3 & 6) lead to additional accumulation on the avalanche

deposits (3) while the removal of snow in the release areas locally reduces the surface mass balance (1). Avalanches from outside (4-5) the glacier boundaries, from seasonally snow-covered slopes or other glaciers, lead to a local increase in mass balance.

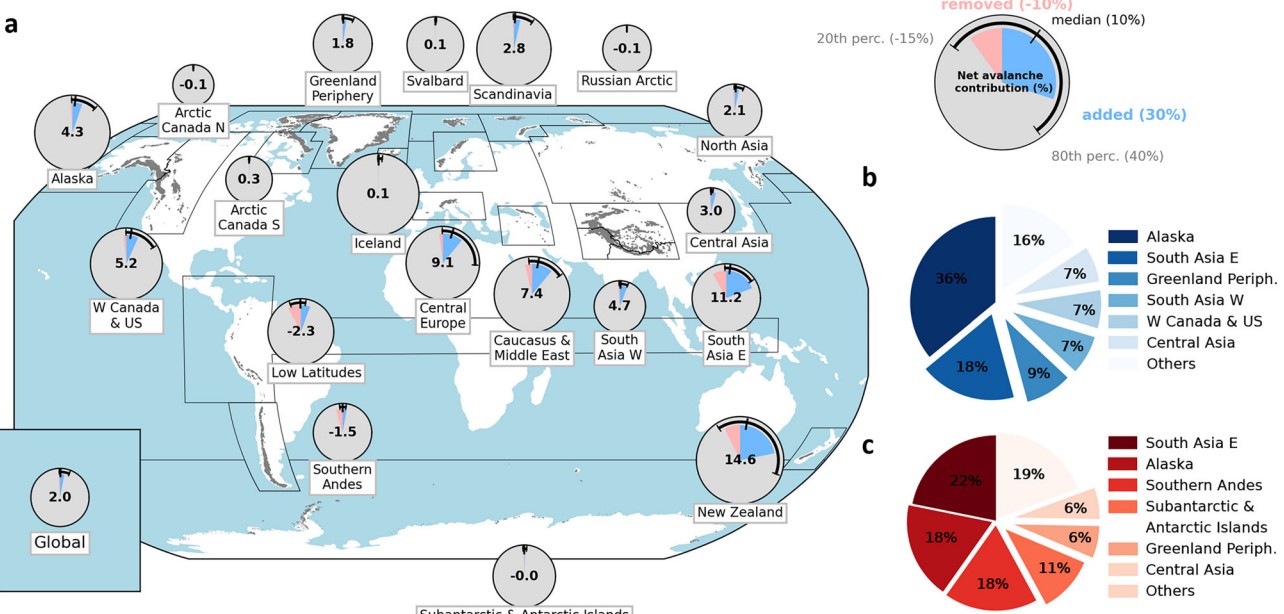

**Fig. 2 | Regional-scale avalanche contribution to glacier accumulation. a** Mean avalanche contribution to glacier accumulation over the period 01/2000-12/2019 for each glacierized region on Earth along with the median, 20th and 80th percentile values of the glacier-wide avalanche contribution to accumulation for all glaciers of the different regions. The size of each pie chart is scaled to the regional accumulation (in m w.eq. yr⁻¹), and the number at the center of each pie chart indicates the regional net avalanche contribution. Partitioning of the global positive (**b**) and negative (**c**) on-glacier avalanche mass contribution between regions. RGI 6.0 regions and glacier outlines (in grey) from[39], which is published under a CCBY license (https://creativecommons.org/licenses/by/4.0/), with no changes made.

0.49 m w.eq. yr⁻¹, followed by South-East Asia (11.2%; 0.23 m w.eq. yr⁻¹) and Central Europe (9.1%; 0.22 m w.eq. yr⁻¹). The regions with an avalanche contribution close to 0%, the Arctic Canada North and South, Svalbard, Iceland, the Russian Arctic and the Subantarctic and Antarctic Islands, are characterized by relatively flat glaciers and ice caps with few avalanche-prone slopes[40]. At the global scale, we estimate that avalanches contribute 2.0% of the snow accumulation on glaciers, corresponding to 0.03 m w.eq. yr⁻¹, and 20% of the world's glaciers have more than 5.9% of their accumulation coming from avalanches. In total, avalanches add on average ~29,000 Mt yr⁻¹ of snow on glaciers and remove ~11,000 Mt yr⁻¹. Alaska and South Asia East have the highest total positive and negative on-glacier avalanche contributions of all glacierized regions on Earth, together accounting for more than half of the global positive and nearly one third of the global negative avalanche contributions (Fig. 2b-c; Supplementary Table 1). The Low Latitudes and Southern Andes regions are the only regions with a negative regional avalanche contribution (−2.3%; −0.04 m w.eq. yr⁻¹ and −1.5%; −0.02 m w.eq. yr⁻¹), and 18% of the snow removed by avalanches from glaciers comes from the Low Latitudes despite the relatively small glacier area of the region. These numbers highlight the importance of this gravitational snow removal process for glaciers in the Andes.

The avalanche contribution is smaller for large glaciers (Supplementary Fig. 1), as avalanches tend to only affect the mass balance at the margins of extensive and relatively flat glacier tongues[17]. Glaciers with steeper average slopes tend to be more influenced by avalanches, whether by snow addition or removal, while glaciers with a positive avalanche contribution tend to have a higher percentage of debris cover (Supplementary Figs. 2–4). This confirms past observations conducted at the scale of High Mountain Asia[41], which likely indicate that debris-covered and avalanche-fed glaciers exist in a similar topographic setting, with steep surrounding slopes. The glacier-wide net avalanche contribution is also negatively correlated with the glacier mean elevation (Supplementary Figs. 2–4). Therefore, avalanches appear to be contributing to maintaining glaciers at lower elevations for a given climate[10,19,42] (Supplementary Fig 2-4).

## Increased spatial variability of glacier mass balance with avalanches

Glacier models implicitly account for avalanches because they are calibrated against geodetic mass balance observations, which integrate the effects of all the different processes affecting the mass balance[26]. However, the spatial variability of accumulation introduced by local avalanche inputs at the glacier scale is not represented, and their impact on projections of future glacier mass changes has never been estimated. Here, we couple our gravitational snow redistribution model to the Open Global Glacier Model (OGGM)[27], to explicitly account for avalanches. We then perform two sets of simulations: the 'Control' simulations, which correspond to the OGGM simulations without explicit representation of avalanches, and the 'Avalanches' simulations, where avalanches are represented using our gravitational snow redistribution model. For the 'Avalanches' simulations, we use the distributed avalanche correction factors calculated over 01/2000-12/2019 to adjust the initial precipitation along the glacier flowline before recalibrating the model parameters against the geodetic mass balance data from ref. [2] (Methods). We note that we did not update the avalanche correction factors over time (Methods). For the recalibration we use an informed three-steps calibration procedure introduced in OGGM v1.6[43], starting with the recalibration of the precipitation correction factor. Given that avalanches tend to add mass to glaciers at the regional scale (Fig. 2), accounting for avalanches generally leads to lower precipitation correction values, with mean regional differences up to 0.1 for Central Europe (Supplementary Fig. 5). This is especially the case in regions with steep relief, which are more prone to avalanching (Fig. 2).

The glacier-wide mass balance is preserved given that we recalibrate against the 01/2000-12/2019 geodetic mass balance from ref. [2]. However, accounting for avalanches increases the spatial variability in the modelled mass balance of individual glaciers, which may also affect the mass balance in avalanche-free elevation bands due to the recalibration procedure (Fig. 3). These changes can produce different volume change trajectories between the 'Control' and 'Avalanches' simulations (Fig. 3). Argentière Glacier in the French Alps accumulates

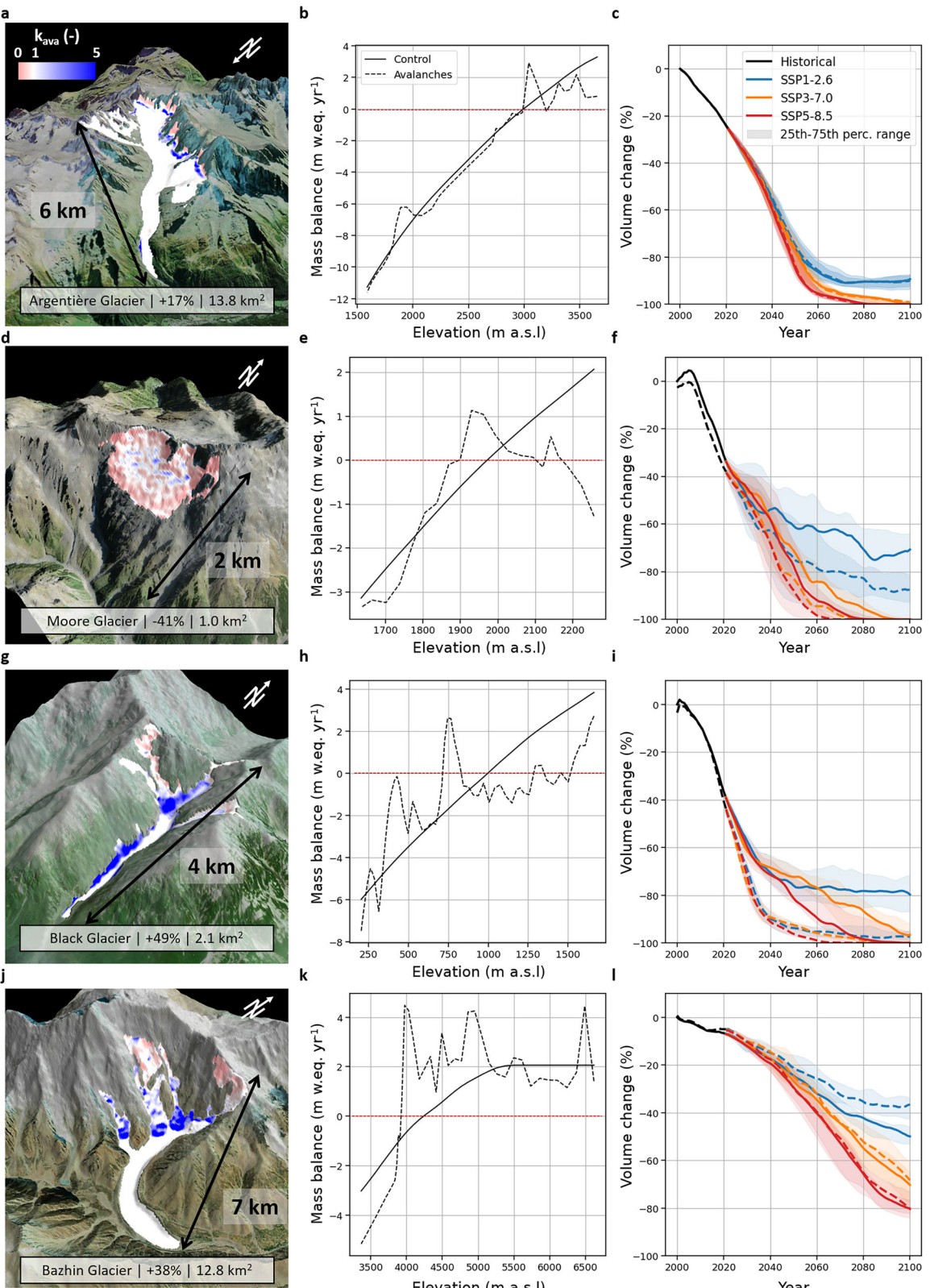

a considerable amount of mass by avalanching at the base of steep north-facing headwalls (Fig. 3a)[21], and our modelled avalanche contribution of +17% agrees well with these remote sensing-based estimates. Our model shows that snow is redistributed from the upper headwalls onto the glacier surface in the accumulation area, and further down in the ablation area (Fig. 3a), with the recalibration leading to slightly more negative mass balance on the main glacier tongue

(Fig. 3b). Despite the substantial contribution of avalanches to the accumulation there are limited differences in the future glacier volume change between the simulations with and without explicit representation of avalanches, which is consistent with the higher order ice flow simulations conducted by ref. 21.

Explicitly accounting for avalanches can have more effect on other glaciers and considerably change both their mass balance

**Fig. 3 | Effect of avalanches on the mass balance and evolution of individual glaciers. a–c** Argentière Glacier in Central Europe, **d–f** Moore Glacier in New Zealand, **g–i** Black Glacier in Alaska, and **j–l** Bazhin Glacier in South Asia West. The left panels show a 3D view of the glaciers with the average avalanche correction factor ($k_{ava}$) over the period 01/2000-12/2019 indicated by the red-white-blue shade wrapped over the glacier extents. The glacier extents are from the Randolph Glacier Inventory (RGI) 6.0[39], which is published under a CCBY license (https://creativecommons.org/licenses/by/4.0/), with no changes made. The background topography is from the DEM used to run the OGGM simulations. The numbers indicate the glacier-wide avalanche contribution and the glacier area. The central panels show the altitudinal mass balance for the 'Control' and 'Avalanches' simulations, with the red dashed line corresponding to a mass balance value of 0 m w.eq. yr[-1]. The right panels show the past and future volume changes of these glaciers as modelled using OGGM. All percentages are given relative to the initial glacier volume in 2000 in the Control scenario. The black line corresponds to the historical period over which the mass balance model was calibrated using W5E5v2.0 data[33]. The colored lines show the median future projections for different SSP scenarios, and the shaded areas indicate the 25th-75th percentile range. The different curves were smoothed using a 5-year rolling mean. The precise location of the glaciers is shown in Supplementary Fig. 6.

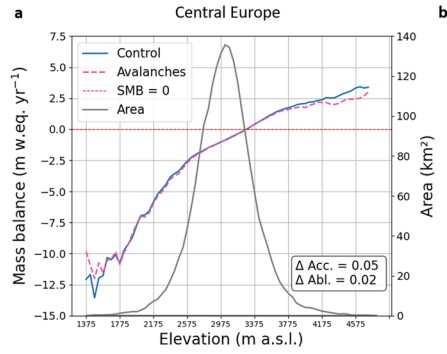

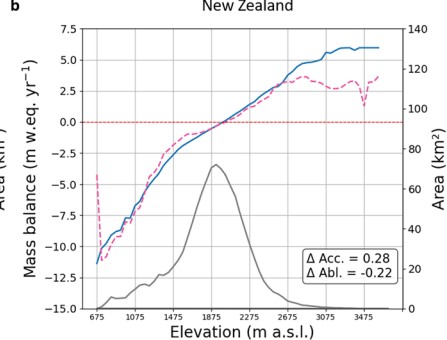

**Fig. 4 | Influence of avalanches on the regional mass balance altitudinal profiles of Central Europe and New Zealand.** Altitudinal mass balance profiles with (blue) and without (pink) avalanches for Central Europe (**a**) and New Zealand (**b**). These profiles were obtained by taking the area-weighted average of the altitudinal profiles of all glaciers of the corresponding regions over the period 01/2000-12/2019. The gray line shows the glacier hypsometry and the red dashed line corresponds to a mass balance value of 0 m w.eq. yr[-1]. The values in the lower right corner indicate the area-weighted mean difference between the mass balance with and without avalanches in the accumulation and the ablation zones.

altitudinal patterns and their future volume changes. An example is Moore Glacier in New Zealand, located on a steep south-west facing slope (Fig. 3d–f). Most of the snow falling both in the accumulation and in the lower ablation zone of this glacier is redistributed to lower elevations including outside of the glacier, leading to 41% of the snow falling on this glacier to be removed by avalanches. This translates into a close-to-balance mass budget at the very top of the glacier, and a positive mass balance at the current glacier median-elevation (Fig. 3e, f). Accounting for this effect results in faster glacier retreat, with the glacier maintaining 57% less volume in the 'Avalanches' simulations than in the 'Control' simulations by the end of the century for the low emission (SSP1-2.6) scenarios (Fig. 3f). For the Black and Bazhin Glaciers in Alaska and South Asia West, a high glacier-wide avalanche contribution of +49% and +38%, respectively, translates in completely different behaviours (Fig. 3g–l). Indeed, on the one hand Black Glacier retreats much faster and reaches a volum.e 87% lower than in the 'Control' simulations by 2100 for SSP1-2.6, while Bazhin Glacier maintains 27% more volume. These differences in glacier projections for a similar glacier-wide avalanche contribution come from the local topography which can result in some cases in snow being redistributed from the accumulation zone into the ablation zone, thereby enhancing mass loss, and in other cases in the expansion of the accumulation zone by increasing the mass supply from higher elevation (Fig. 1). This also implies that while small avalanche-fed glaciers may be temporarily sheltered from atmospheric warming, they could also become very sensitive to it if rising temperatures affect the avalanche supply[20].

At the regional scale, explicitly accounting for avalanches reduces mass accumulation at high elevations, while the snow redistributed to lower elevations leads to less negative mass balances (Fig. 4; Supplementary Fig. 7). The area-weighted mean mass balance is reduced by 0.05 m w.eq. yr[-1] in the accumulation zone of Central European glaciers, and by 0.28 m w.eq. yr[-1] in New Zealand (Fig. 4). Lower accumulation values are also visible in the other strongly avalanche-

affected regions, such as the Caucasus and Middle East (0.11 m w.eq. yr[-1] reduction), South Asia East (0.25) or the Low Latitudes (0.14), while the other regions are less affected (Supplementary Fig. 7). This reduction is most noticeable at the highest elevations, which also have a smaller area, so the effect on the glacier-wide mass balance is limited. The generally lower mass balance at high elevations in the strongly avalanche-affected regions highlights the sensitivity of the steep glaciers and glacierized headwalls to rising air temperatures. Indeed, these high-elevation glaciers have a low mass turnover because most of the deposited snow in these zones is gravitationally redistributed to lower elevations (Fig. 3). This enhanced sensitivity to warming at high elevations, highlighted by our regional-scale simulations, has been observed locally - for example in the Mont-Blanc massif, where retreating ice aprons are linked to increasing instability of steep mountain headwalls[44–46]. At the glacier scale, reduced mass balance estimates observed when accounting for gravitational snow redistribution therefore indicate locations where glaciers are likely to shrink rapidly at high elevations, resulting in potential headwall instabilities due to the rapid exposure of permafrost (Fig. 3).

## Longer subsistence of small glaciers with avalanches
Explicitly representing avalanches in OGGM has little effect on the regional-scale simulations of glacier volume and area changes from 2000 to 2100 (Fig. 5; Supplementary Figs. 8–25), with differences not exceeding 2%, even in the most avalanche-affected regions (Supplementary Fig. 8–25). This result arises because the mass balance model with or without avalanches is calibrated against the same geodetic mass balance data[2,4,6], and that the largest glaciers - generally less avalanche-affected (Fig. 2; Supplementary Fig. 1) - contribute the most to the regional volume change projections. Similar observations have been made in previous efforts to include sub-debris melt or calving in global glacier models[6,8,30].

However, given the influence of the avalanches on the altitudinal mass balance patterns, explicitly accounting for them can have a

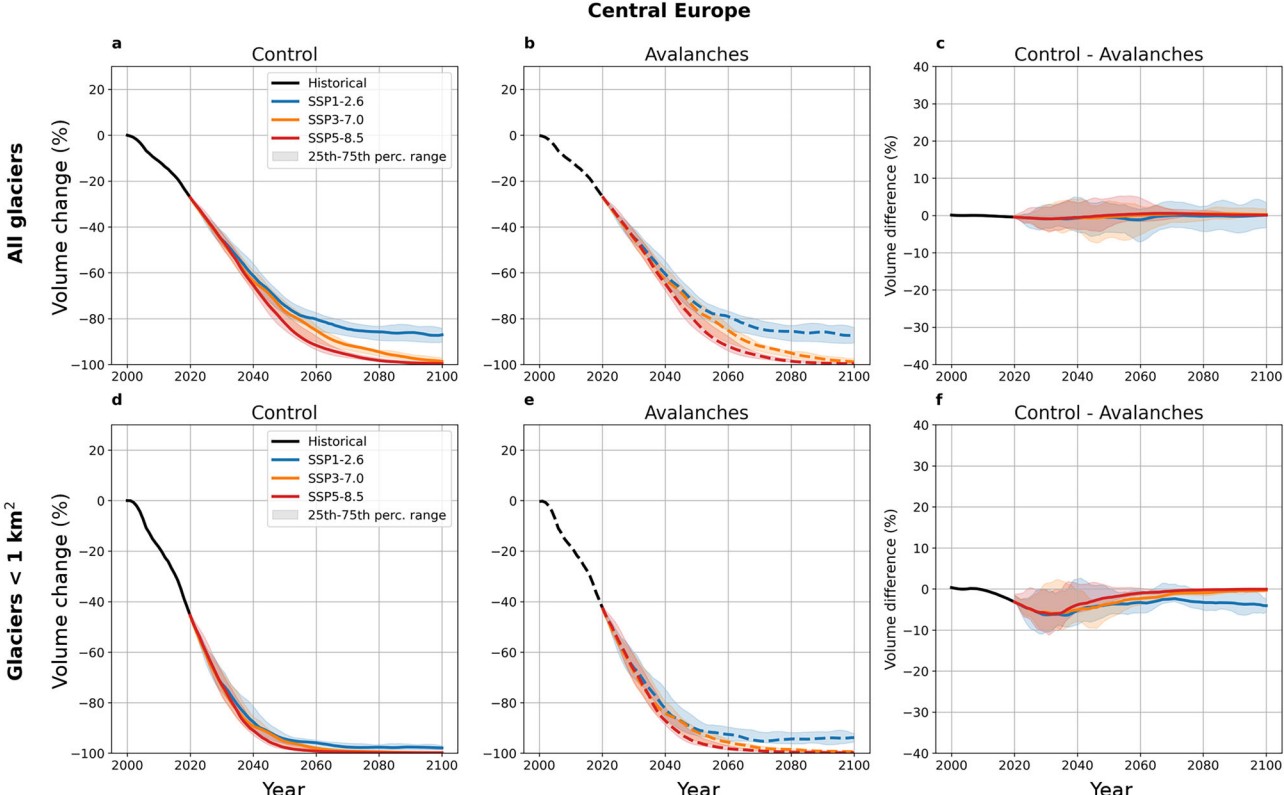

**Fig. 5 | Influence of avalanches on regional glacier volume changes in Central Europe.** Projected volume changes of all glaciers (**a**–**c**) and only the glaciers smaller than 1 km² (**d**–**f**) in Central Europe, from the 'Control' (**a**, **d**) and 'Avalanches' (**b**, **e**) simulations. The right-hand plots (**c**, **f**) show the difference between the 'Control' and the 'Avalanches' simulations, so that negative values indicate more volume in the 'Avalanches' simulations. All percentages are given relative to the initial volume in 2000 in the 'Control' simulation. The black line corresponds to the historical period over which the mass balance model was calibrated using W5E5v2.0 data[33]. The colored lines show the median future projections for different SSP scenarios, and the shaded areas indicate the 25th-75th percentile range. The different curves were smoothed using a 5-year rolling mean.

considerable effect at the scale of individual glaciers (Figs. 3, 5). When examining glaciers smaller than 1 km² in the different RGI regions most affected by avalanches (Fig. 5; Supplementary Figs. 7–25), we find that accounting for avalanches leads to a slower volume loss than previously anticipated. Notably, the difference between the 'Control' and 'Avalanches' simulations peaks between 2020 and 2050, depending on the region and the SSP scenario. Furthermore, while volume losses under SSP3-7.0 and SSP5-8.5 are nearly identical by the end of the century with and without accounting for avalanches, in some regions small glaciers reach a different equilibrium under SSP1-2.6 with more volume preserved than when not accounting for avalanches.

In Central Europe, explicitly accounting for avalanches under SSP1-2.6 leads glaciers smaller than 1 km² to retain 0.98 km³ of ice - three times more than in the 'Control' simulations (Fig. 5). For glaciers smaller than 0.5 km³, four times more ice is retained. In South Asia West, glaciers smaller than 1 km² (0.5 km²) retain 19% (23%) more ice in the 'Avalanches' simulations (Supplementary Fig. 20). These results confirm the hypothesis of local-scale studies predicting the rapid and complete disappearance of small low-elevation glaciers, except for those strongly fed by avalanches[12,15,47]. Here, we provide a first estimate of this effect across all mountain regions, which is important to consider given the role of small glaciers in downstream hydrology of mountain catchments[48].

### Changing influence of avalanches as glaciers retreat
As glaciers retreat in a warming climate, the avalanche contribution is expected to be influenced by deglaciation of avalanche release and deposit slopes, glacier surface lowering, and changes in snowfall amounts including through the rise of the rain-snow transition line[49]

(Methods). The avalanche contribution may increase or decrease depending on mountain topography and future climate projections (Fig. 6, Supplementary Figs. 26–41). In Central Europe, glaciers are anticipated to retreat to the base of steep headwalls under all SSP scenarios[12,47] (Fig. 6a), increasing the avalanche contribution from 11% in 2020 to 13–15% by 2100 (Fig. 6b). In New Zealand, the avalanche contribution depends strongly on the SSP scenario: it is expected to remain at its current level under SSP1-2.6, but decrease in higher emission scenarios, from 15% in 2020 to 3% in 2100 under SSP5-8.5 as the glacierized area shrinks (Fig. 6d–f). This likely reflects a dependence on the rising snow-line which, in high-warming scenarios, would rise above most of the relatively low summits in this range. In the Low Latitudes, glaciers will retreat to steeper, high-elevation slopes, causing relatively more accumulation to be removed by avalanches. As a result, the avalanche contribution is projected to decrease from −4% in 2020 to −10% in 2100 under SSP1-2.6 and from −4% to −30% under SSP5-8.5 (Fig. 6g–i). Changes in relative accumulation (in m w.eq. yr⁻¹) appear to have a limited influence on the avalanche contribution, as these values remain consistent over time and across SSP scenarios in the different regions (Fig. 6c, f, i).

These simulations of avalanche contribution strongly depend on the topography of the glaciers and their surrounding catchments, and on the evolution of the glacier extents as simulated by our model. OGGM is a flowline model that represents the glacier evolution by elevation band (Methods) and requires an entire elevation band to be ice-free before removing it from the glacier extent, which can sometimes lead to inaccurate representations of the glacier geometry. Furthermore, while we are confident that our model correctly depicts the general regional trends, we note that in these simulations the

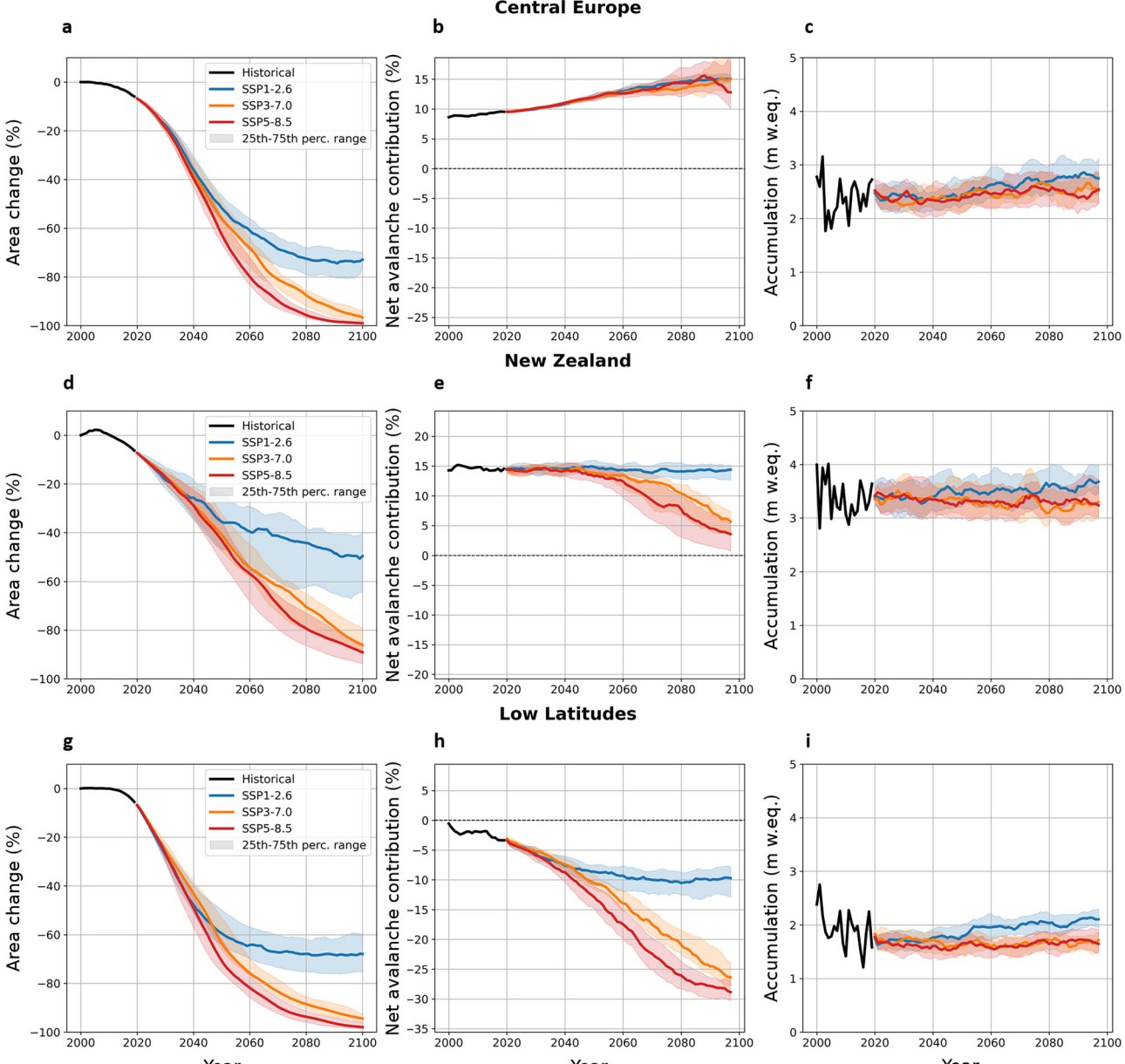

**Fig. 6 | Future evolution of regional avalanche contribution.** Projected changes of glacier area (**a**, **d**, **g**), avalanche contribution to accumulation (**b**, **e**, **h**) and total snow accumulation (**c**, **f**, **i**) for Central Europe (**a–c**), New Zealand (**d–f**) and the Low Latitudes (**g–i**), in the 'Avalanches' scenario. The scales are identical between the panels but for the avalanche contribution the 0% line marked by a black dashed horizontal line changes position. The black line corresponds to the historical period over which the mass balance model was calibrated using W5E5v2.0 data[33]. The colored lines show the median future projections for different SSP scenarios, and the shaded areas indicate the 25th–75th percentile range. The different curves were smoothed using a 5-year rolling mean, except for the historical accumulation.

avalanche contribution at the end of the century is dependent on the accurate geometric representation of the few glaciers remaining by 2100 in the regions most affected by glacier retreat. As a result, the uncertainty of these simulations is likely higher at the end of the century than the currently depicted range, which does not include uncertainties related to the glacier geometry. Despite these limitations, the contrasting regional patterns highlight the need to account for avalanches on individual glaciers, as overall glacier shrinkage[6] will likely increase the role of these processes in the mass balance of glaciers across most mountain ranges[47].

### Implications for modelling the evolution of glaciers
This study provides the first global estimate of the contribution of avalanches to glacier accumulation. We estimate that 3.0% of the snow that falls on glaciers originates from avalanches, while 1.1% of

the accumulated snow is removed by them. The importance of this process varies strongly across mountain ranges and individual glaciers. In regions such as New Zealand, High Mountain Asia, Central Europe, the Andes, Caucasus, Scandinavia, Alaska and Western Canada and USA, avalanches play a key role for the glacier mass balance and are anticipated to become increasingly important as glaciers retreat under a warming climate[16,21,50]. Accounting for gravitational snow redistribution in glacier models alters the altitudinal mass balance patterns and, in turn, affects projections of the evolution of individual glaciers - especially the smaller ones which are relatively more avalanche-prone. While the effect on regional-scale ice volume projections is limited, the fate of these small ice masses has strong implications for landscape evolution and local hydrology[12]. This study also paves the way for more detailed sub-regional studies, for example examining links between avalanches

and debris cover (Supplementary Figs. 2–3), differences in avalanche effects across glacier types (valley, cirque), or the impacts of avalanches occurring in accumulation versus ablation zones (Fig. 3).

Our modelling approach is sensitive to the spatial resolution of the DEMs used and the temporal resolution at which the avalanche contribution is updated, though these factors have only limited influence on our results (Methods; Supplementary Figs. 42–48; Supplementary Table 3). More importantly, because OGGM is a flowline-based model, avalanche contributions must be averaged by elevation band[27]. This averaging smooths out the highly localized nature of avalanche inputs[13,21] and may obscure important spatial variability. Indeed, localized avalanche inputs are expected to affect glacier dynamics, especially as glaciers retreat in a warming climate, leaving isolated ice patches at the base of steep headwalls[21]. Recent advances in deep learning–based ice flow models offer promising avenues for distributed simulations at the global scale and should enable a better representation of such localized processes[51,52]. A major challenge, however, lies in calibrating and validating gravitational snow redistribution models in glacierized areas due to the scarcity of remote sensing and field-based observations. Expanding such datasets will be crucial to refine future model estimates (Methods), particularly to capture feedbacks related to avalanche deposition, such as effects on albedo and snow density, and to better understand how avalanche activity depends on the characteristics of release areas (aspect, slope, or hanging glaciers).

Despite these challenges, our approach provides a first-order estimate of the current and future contribution of avalanches to the mass balance of every individual glacier across the world and evaluates the impact on the projections of their future evolution. Our results underscore the need for an improved representation of local processes affecting the spatial variability of mass balance in glacier models, which, beyond avalanches, include frontal ablation of marine- and lake-terminating glaciers, variable melt under debris, variable precipitation with elevation or snow redistribution by wind. Further research is needed to understand how these processes interact[30,32,38,53] and how they can be effectively integrated into global glacier models[6,8,30]. For example, wind-driven snow redistribution may locally enhance or reduce the effective snow depth on slopes on or in the vicinity of glaciers, and therefore affect the avalanche supply. These process representations, combined with the recent developments in remote sensing data availability and assimilation by global glacier models and the anticipated capability of running these simulations in a distributed way, will pave the way for a new generation of models capable of accurately projecting the future of every individual glacier across Earth's mountain ranges.

## Methods

### Avalanche model description
Our avalanche model is based on the SnowSlide model[31], which has been widely used in glacio-hydrological studies of individual catchments[18,54–58]. This model assumes an exponential decay of the maximum snow depth as a function of slope:

$$SND_{max} = Ce^{-\alpha S} \tag{1}$$

Where $SND_{max}$ is the maximum snow depth at any given point of the digital elevation model (DEM) grid, $S$ is the slope of the snow DEM (DEM with the snow layer on top), and $C$ and $\alpha$ are the two parameters used to define this maximum snow depth, along with the minimum slope of 25° over which redistribution is allowed to occur[31]. Once the maximum snow depth is defined, the excess snow is routed recursively using the multiple flow direction routing algorithm[31,59] until no snow is redistributed anymore.

The large majority of studies that have applied this model have done it at hourly time-steps and using the original parameters proposed by ref. 31, originally obtained based on the comparison between end-of-season snow extents from Landsat ETM+ satellite images over the Waltzmann massif and the outputs of the SnowSlide model run at hourly time-steps and 30 m spatial resolution. More recently[32], used the same maximum snow depth parametrization than[31] at 25, 50 and 100 m resolution, but only using the minimum slope parameter of 25° for triggering, and reducing by 30% the maximum snow depth for pixels receiving snow, to mimic avalanche dynamics. They validated this approach against snow depth maps from repeat LiDAR acquisitions in the eastern Swiss Alps, which indicated a good agreement with the modelled snow depths.

For this study, we use values of 0.14 and 145 for the parameters $\alpha$ and $C$, respectively, along with the same parameterization of avalanche inertia introduced by ref. 32. We did not use a minimum slope threshold as we did not see any difference when including it. The values of $\alpha$ and $C$ differ slightly from those originally reported by ref. 31 – 0.12 and 120, respectively. They are obtained based on a qualitative assessment of SnowSlide outputs in the Mt Blanc massif, and Argentière Glacier specifically, where we could directly compare our model outputs with distributed mass balance data from an ice flux inversion[21]. This parametrization was further evaluated against Sentinel-1 avalanche deposits (Methods - Avalanche model evaluation).

We run the snow redistribution model at monthly time-steps and at a spatial resolution $\Delta x$ (in m), varying as a function of glacier size $S$ (in km²)[27]:

$$\Delta x = min(d_1 \sqrt{S} + d_2, 200) \tag{2}$$

Where the parameters $d_1$ and $d_2$ are set to 14 and 10, respectively. If the chosen spatial resolution is larger than 200 m, we clip it to this value, so the spatial resolution effectively ranges between 10 and 200 m. We tested the sensitivity of our results to the spatial resolution below.

### Avalanche correction factors
We run SnowSlide independently for every month of the year using the 01/2000-12/2019 mean-monthly precipitation from W5E5v2.0 data[33] as inputs, along with a density of 200 kg m⁻³ to convert the solid precipitation inputs to a snow height. SnowSlide is run over the whole glacier and its surroundings within a buffer of 80 grid cells (with a spatial resolution varying from 10 m to 200 m) around the glaciers, which we assume to encompass the entire glacier catchment. We then sum the monthly snow maps and take the ratio of snow accumulation before and after redistribution to compute a distributed avalanche correction factor $k_{ava}$. The values within the glacier outlines from the Randolph Glacier Inventory (RGI) version 6.0[39] are then aggregated:

1. Over the entire glacier to obtain a glacier-wide avalanche correction factor $k_{ava,gl}$.
2. Along the glacier flowline to obtain an avalanche correction factor for each elevation band $k_{ava,i}$ following[60].

We note that the use of RGI 6.0 outlines, for example due to potential misalignments, can lead to uncertainties in glacier future projections and also most likely in estimates of the avalanche contribution[61]. At the regional scale the uncertainties are small relative to other contributions such as from the choice of climate models[62], but for individual glaciers these estimates should be interpreted with caution.

### Recalibration of the glacier model
The avalanche correction factors $k_{ava,i}$ vary along the glacier flowline and can be applied for each glacier as multiplicative factors to the monthly solid precipitation $P_i^{Solid}(z)$ (in kg m⁻² month⁻¹) to compute the mean monthly mass balance $B_i(z)$ using the temperature index

model from OGGM building from[27,29,63] and expressed as:

$$B_i(z) = k_{ava,\,i}(z) \times k_p \times P_i^{Solid}(z) - d_f \times max(t_b + T_i(z) - T_{Melt}, 0) \quad (3)$$

Where $T_i(z)$ is the mean monthly air temperature (in °C) at altitude $z$ using a −6.5 °C km$^{-1}$ lapse rate and $T_{Melt}$ is the monthly mean air temperature above which ice melt is assumed to occur (−1 °C per default). In OGGM precipitation is directly taken from the nearest climate reanalysis cell, without any correction for elevation, contrary to the temperature. The precipitation is considered to fall in solid (liquid) form when the mean monthly temperature is below 0 °C (above 2 °C), with a linear transition in-between.

The precipitation correction factor $k_p$ (·), the degree-day factor $d_f$ (in kg m$^{-2}$ K$^{-1}$ month$^{-1}$) and the temperature bias $t_b$ (in K) are three parameters that are calibrated for each glacier against the 01/2000-12/2019 average geodetic mass balance from ref. 2 and using W5E5v2.0 data[33]. This calibration uses the informed three-steps approach introduced in OGGM v1.6[43], which is run until the geodetic mass balance is matched following these steps:

1. the three parameters are prescribed some allowed ranges ($k_p \in [0.1, 10]$, $d_f \in [0.33, 33]$ kg m$^{-2}$ K$^{-1}$ day$^{-1}$, $t_b \in [-8, 8]$ K),
2. $k_p$ is obtained from the glacier's winter precipitation from W5E5v2.0 based on a logarithmic relation derived by ref. 29,
3. $d_f$ is fixed at 5 kg m$^{-2}$ K$^{-1}$ day$^{-1}$, $t_b$ is calibrated to match the geodetic mass balance of each glacier, and its median value per W5E5v2.0 grid point is attributed to each glacier in the grid point,
4. $k_p$ is recalibrated allowing it to vary between +/-20% of its original value, then $d_f$ and then $t_b$ are recalibrated.

When accounting for avalanching, we directly use the W5E5v2.0 monthly solid precipitation multiplied by the avalanche correction factor ($k_{ava,\,i} \times P_i^{Solid}(z)$) and the pre-calibrated $k_p$, $d_f$ and $t_b$. We then recalibrated these parameters starting from step 4. In practice, introducing avalanches has two main interconnected effects after calibration: first, accumulation patterns along the glacier are modified according to the avalanche model; second, the initial precipitation correction factor, originally estimated without accounting for avalanche contributions, is recomputed based on this new information. In the cases where no initial precipitation factor can be found within the specified +/−20% bounds, $d_f$ and $t_b$ are changing as well, as described in step 4.

Following these different steps, we obtain two mass balance models, both calibrated against the 01/2000-12/2019 geodetic mass balance for all glaciers in the world: the 'Control' model which does not explicitly account for avalanches, and the 'Avalanches' model, where avalanche contribution is explicitly accounted for using the $k_{ava,\,i}$ avalanche correction factors.

## Avalanche contribution to glacier accumulation

We compute the net glacier-wide avalanche contribution to accumulation $Av_{gl}$, expressed as a percentage:

$$Av_{gl} = \frac{SND_{with\,ava,gl} - SND_{no\,ava,gl}}{SND_{with\,ava,gl}} \times 100 \quad (4)$$

Where $SND_{no\,ava,gl}$ and $SND_{with\,ava,gl}$ are the snow depths before and after avalanche redistribution, integrated over the entire glacier scale. This net glacier-wide avalanche contribution can be split between the positive avalanche contribution where snow is added to the glacier ($Av_{gl}^+$), and the negative avalanche contribution where snow is removed from the glacier ($Av_{gl}^-$):

$$Av_{gl} = Av_{gl}^+ + Av_{gl}^- \quad (5)$$

Similarly, we compute the net region-wide avalanche contribution for each of the 19 RGI regions and for the entire globe by summing the snow accumulation before and after avalanches for all glaciers of the regions:

$$Av_{reg} = \frac{\sum\limits_{glaciers} (SND_{with\,ava,gl} - SND_{no\,ava,gl})}{\sum\limits_{glaciers} SND_{with\,ava,gl}} \times 100 \quad (6)$$

## Comparison to glacier metrics

We compare the net, positive and negative glacier-wide mean avalanche contributions calculated over the period 01/2000-12/2019 with a series of glacier metrics. The average slope along the flowline, size and mean elevation are derived from the RGI 6.0 glacier attributes[39], the geodetic mass balance is obtained from the dataset by ref. 2 and the percentage of debris cover is obtained from the global dataset by ref. 64.

## Glacier projections

We simulate the evolution of all glaciers in the world for the period 2000–2100 with both the 'Control' and the 'Avalanches' models. As climate data for 2020–2100 we use all the available models out of the 18 climate models of the Coupled Model Intercomparison Project (CMIP) Phase 6 for the three different Shared Socioeconomic Pathways (SSP1-2.6, SSP3-7.0 and SSP5–8.5). These climate data are downscaled and bias-corrected to the W5E5 data following the procedure described in ref. 65. We retain all the glaciers for which at least 75% of the SSP/GCM combinations ran without errors, which corresponds to 98.6% of the RGI 6.0 glaciers (97.7% of the total glacier area; Supplementary Table 2). For these projections we take the avalanche correction factor calculated over the period 01/2000-12/2019 and do not update it with time, because of the high associated computational costs. We did however test the influence of updating this avalanche correction factor every 20 years for 1255 glaciers with a glacier-wide avalanche correction factor higher than 1.2 or lower than 0.8 (corresponding to an avalanche contribution higher than 17% or lower than -25%), randomly selected in 15 of the RGI regions for the ipsl-cm6a-lr_r1i1p1f1 GCM under SSP1-2.6 (Supplementary Figs. 42–43). We found the differences between the avalanche scenarios with or without temporal update of the avalanche correction factor to be in most cases negligible relative to the differences between the avalanche and the control scenarios. This suggests that although glacier surface lowering (creating steeper and taller headwalls) and a rising rain-snow transition may affect avalanche-driven mass redistribution in the future[20,49], these influences are likely secondary compared to the main climatic controls on glacier evolution. In the few cases where the temporal update of avalanches made a noticeable difference like for the glaciers with a positive avalanche contribution in Alaska or North Asia (Supplementary Fig. 43), it increased the gap between the control and avalanche scenarios. This suggests that not updating the avalanche contribution, as done in this study, gives a lower-bound estimate of its long-term influence on glacier evolution.

## Avalanche model evaluation

While SnowSlide has been extensively used and validated, and is considered to give a realistic estimate of avalanche redistribution for hydrological models[18,21,31,56–59,66], we extend these validation efforts to glacierized regions using remote sensing observations.

We validate our avalanche model against avalanche deposit extents derived from Sentinel-1 images over a 5-year period for the Mt Blanc, Everest and Hispar regions[17]. While the data derived from Sentinel-1 is limited to deposit extents and does not give any information on redistributed volume or mass, we show that the deposits from our model occur at the same elevation range as the deposits

derived from Sentinel-1 images, albeit covering a generally smaller area, especially in the Hispar region (Supplementary Figs. 49–51). These results also highlight a strong correlation between the observed and modelled avalanche deposit areas for individual glaciers ($R^2 \geq 0.73$), which confirms that our model adequately represents the spatial domain of the observed avalanche activity.

Direct measurements of avalanche contribution to glacier accumulation would be more useful but it is notoriously difficult to measure from remote sensing or field data[13,21]. For this, we mostly rely on the validation efforts by ref. 32, although they were conducted for non-glacierized regions. Some studies have estimated the local contribution of avalanches to glacier mass balance using Ground Penetrating Radar measurements[13,20], but highlighted a strong interannual and spatial variability, making it challenging to attribute the variability to avalanches and to obtain a glacier-wide contribution estimate. We compared the modelled glacier-wide precipitation correction factor with the value obtained by ref. 21 from an inversion of the ice flux constrained by remote sensing observations for Argentière Glacier, finding a good agreement between the two with the measured and modelled avalanche correction factor $k_{ava,gl}$ being equal to 1.2 and 1.3, respectively, for the glacier without its North tributary, and 1.1 and 1.3 with all the tributaries.

Ultimately, we compared the modelled and measured annual mass balance for all measurements available over the period 01/2000-12/2019 in the global database from the World Glacier Monitoring Service[67]. 9646 annual point mass balance measurements were retained, acquired across 90 glaciers distributed across all RGI regions except Arctic Canada South, the Russian Arctic, North Asia, the Caucasus and Middle East, South Asia West and the Subantarctic and Antarctic Islands. Including avalanches in the mass balance model had a negligible impact on the comparison with the mass balance measurements, with a RMSE of 1.15 m w.eq. yr$^{-1}$ for the Control scenario and 1.17 m w.eq. yr$^{-1}$ for the Avalanches scenario, and the same $R^2$ (Supplementary Fig. 52). This is not surprising given that due to representativity and safety considerations, the huge majority of mass balance measurements are conducted away from any avalanche deposits and the long-term monitoring sites have a limited estimated avalanche contribution, with 90% of these glaciers having an avalanche contribution between −2% and +10% (Supplementary Fig. 53).

### Sensitivity of the model to DEM spatial resolution

Given the computational constraints of simulating snow redistribution on all glaciers of the world and surrounding catchments, we use a spatial resolution that varies with glacier size, between 10 and 200 m. Coarser spatial resolutions translate into less steep slopes[68], and ultimately in less snow being redistributed by our gravitational snow redistribution model[32]. In the mountainous region of South-East Switzerland[32], showed that SnowSlide simulations at 25 m resolution resulted in the best agreement with LiDAR measurements of snow volumes, but they still found similar snow redistribution patterns when running their simulations at 50 or 100 m, with significant improvement from the simulations without gravitational snow redistribution.

Here, we test the sensitivity of the avalanche contribution and glacier volume changes to the spatial resolution of the DEMs by running our model at 50 m and 100 m resolution for Central Europe (Supplementary Table 3; Supplementary Figs. 44–46). We find the regional avalanche contribution to be similar in all three cases, both for the historical period and for the future simulations, with the results from the reference simulations being more similar to those from the 50 m resolution simulations (Supplementary Figs. 44[46]). Such findings are also true at the scale of small individual glaciers (Supplementary Figs. 47–48). Furthermore, a lower resolution leads to lower avalanche contribution and to a less strong effect of avalanching on the future projections of small glaciers. While this does not influence the findings

of this study, it is worth noting that our variable spatial resolution may lead to an underestimation of the avalanche contribution to larger glaciers relative to the very small ones.

## Data availability
The maps of the distributed avalanche correction factors and the glacier-wide avalanche contributions generated in this study are available on Zenodo at https://doi.org/10.5281/zenodo.17347752[69]. The data used to run the OGGM simulations comes from the OGGM-shop[27].

## Code availability
The code used to simulate avalanches using the SnowSlide algorithm and to couple these simulations to the glacier model from OGGM is available on Github (https://github.com/MarinKneib/Snowslide) and Zenodo[70]. We used OGGM v1.6.2[43] for all the simulations presented here. The 3D maps in Fig. 3 were produced using the Glacier3Dviz library (https://glacier3dviz.oggm.org).

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

## Acknowledgements

This project has received funding from the Swiss National Science Foundation (SNSF) under the Postdoc. Mobility programme, grant agreement P500PN_210739, CAIRN (MK), "Contribution of avalanches to glacier mass balance", and grant agreement P5R5PN_225605, CAIRN-GLOBAL (MK), "Contribution of avalanches to glacier mass balance at the global scale". The authors would like to acknowledge the OGGM community for the extensive online documentation, data resources (OGGM-shop) and computing infrastructure that were used as part of this study.

## Author contributions

Conceptualization: M.K., F.M., F.B.; Methodology: M.K., F.M., G.C.; Software: M.K., F.M., G.C., P.S., L.S.; Data curation: M.K., F.M.; Formal analysis: M.K.; Investigation: M.K., F.M., F.B., G.C., D.F., M.H., M.v.T., A.J., P.S., L.S., A.D., N.C.; Writing – original draft preparation: M.K.; Writing – review and editing: M.K., F.M., F.B., G.C., D.F., M.H., M.v.T., A.J., P.S., L.S., A.D., N.C.; Visualization: M.K., P.S., L.S.; Supervision: M.K., F.M., F.B., D.F.; Funding acquisition: M.K., F.M., F.B.

## Competing interests

The authors declare no competing interests.
