## [Transparent Peer Review file · Nature Communications]

Topographically-controlled contribution of avalanches to glacier mass balance in the 21st century

Corresponding Author: Dr Marin Kneib

Version 0:

Reviewer comments:

Reviewer #1

(Remarks to the Author)

This is a valuable contribution that explores the role of avalanche to glacier mass gain (or loss) at the global scale – the first time such an analysis has been applied to glaciers globally. The authors couple an open source (OGGM) glacier model to a gravitational snow redistribution model (Snow Slide) to explore glacier mass balance with and without modelled avalanche input. Model parameters were calibrated to match the geodetic mass balance developed by Hugonnet et al (2021) and results are also compared to glaciological mass balance data. Future projections were then made for using three SSPs exploring how avalanche accumulation may affect glacier evolution in a warming world. This is a high-quality contribution, it is well written, well cited, and contains a number of quality Figures to support findings. I enjoyed reading this contribution and have only minor comments below.

Specific comments to each section follow:

Title: Perhaps the term 'variable' rather than 'significant' would be better in the title? While the percentage contribution of avalanche input is indeed significant in some glacier regions, many other regions had contributions of less than 1%. When considering Supplementary Fig 6, one might determine that across the global sample the avalanche contribution was not 'significant' but quite significant particularly in alpine settings.

Line 34: warming - minor typo 'warmin'

Fig 1: This is a nice Figure, but the blue lines in 1a are a bit confusing. Does the blue line indicate a hypothetical glacier area?

Line 66: Reference to the 'temperature bias' can this be explained more clearly. Is this referring to a known bias associated with the gridded temperature data, to the known variability of temperature with elevation, or, be a reference to previous reports of elevation dependant warming? A few additional sentences to explain exactly what temperature bias authors are referring would be appreciated.

Line 91: Model simulations have been set-up to align with calendar years. Was there any consideration to using the hydrological year – even though northern and southern hemisphere will be opposing? Starting simulations in January means that the first ablation (accumulation) season is half over in the southern (northern) hemisphere respectively, although equatorial glaciers will be different again. Perhaps this is why the start of the time series look a bit odd in Fig 6 for the NZ region? But maybe this does not matter depending on model spin-up processes?

Line 108 (plus Sup): It is acknowledged that the Randolph Glacier Inventory is the 'standard' for global glacier outlines. However, there exists some rather inaccurate outlines for some glaciers. For example, the RGI outline of Rolleston Glacier (NZ) is at best a poorly aligned representation of the glacier back in the 1980-90s. It is significantly oversize (and misaligned). Since at least 2010 the entire portion of the RGI outline to the south-east is no longer connected to the current glacier, it is well separated by bedrock and not considered part of the glacier. The slopes above this now non-existent part of the glacier are prone to avalanche deposition, so one might expect this will create a positive bias in results. Without in-depth knowledge of all glacier regions, one wonders how many other small glaciers suffer the same poor alignment of RGI outlines, which will consequently impact the accuracy of the geodetic mass balance as produced by Hugonnet et al and

results of this study. Did the authors undertake spot-checks of the glacier areas? I appreciate that this is perhaps beyond the scope of the analysis, but such errors are potentially perpetuating from one global study to the next.

Lines 136-138: The results relating to where on the glacier avalanche deposition occurs is very interesting and ideally warrants further investigation/discussion. Avalanche deposition onto the ablation zone creates a variable range of feedbacks, especially in relation to albedo/melting. Avalanche deposition onto an ablation area may increase or decrease albedo depending on surface type (ice/debris) and avalanche deposition is known to be of higher density, thus more mass. Could the authors say a little more about the differences between glaciers where avalanche deposition is primarily occurring in the accumulation area of glacier compared to avalanche deposition the occurs in glacier ablation areas?

Line 168: It would seem that a number of regions required almost no recalibration of the precipitation factor, which would indicate that at these sites the role of avalanche input (either positive or negative is negligible). Results shown in Sup Fig 5 which are close to null, align with results in Fig 2 for areas where avalanche processes are negligible. One might conclude that in general it is glaciers in mountainous terrain for which avalanche deposition is important. Perhaps this finding could be stated more explicitly.

Line 175-200: The individual glacier case studies are interesting and a great way to tease out specific examples. Could the authors include a Figure (possibly in supplementary) that supports Fig 3 so readers can quickly ascertain exactly where in the regional mountain ranges the various case studies are located.

Line 305-307: As glaciers thin slope angle tends to increase in the headwall region this may be an additional 'future' feedback.

Line 311-315: The Southern Alps of NZ are relatively low elevation compared to mountain ranges around the globe, perhaps this lack of 'real-estate' at higher elevation contributes to the finding that avalanche contribution strongly depends on the SSP. As temperatures warm the snow/rain threshold will likely be off the top of many mountains in NZ.

Lines 350-364: The authors have highlighted some interesting results; this is a great study. However, I think that considering results only in terms of 'geopolitical' regions limits the findings. What are the patterns for cirque glaciers globally compared to valley glaciers (regardless of region)? What about debris-covered glaciers? It appears that the Himalayan glaciers create a bias, I can think of many debris-covered glacier for which avalanche accumulation is not an important contribution. How do glaciers that have significant avalanche deposition in their ablation areas compare to those where avalanche is confirmed to the upper accumulation areas? These are all very interesting spatial questions for which the authors have results, but which are not discussed?

Methods section: This was well written and clear to follow. I found that questions I had on reading the core paper e.g. impact of DEM resolution, role of precipitation factors etc were all answered in due course in this section.

Supplementary: The supplementary is substantial, but all the material is well presented and strongly supports the main document. It was good to see all the regional graphs, although seeing some Figures/results focused on glacier characteristics rather than glacier regions would have been a great addition.

(Remarks on code availability)

I fully support the authors providing the code and this is likely something I will visit and explore in due course, and potentially utilise for post-graduate projects. However, I have not (as yet) used the OGGM, so did not check the code.

Reviewer #2

(Remarks to the Author)

Review of: "Substantial contribution of avalanches to glacier accumulation in the 21st century", M. Kneib et al.

General comment

This paper explores the contribution of gravitational snow redistribution to glacier accumulation, through an extensive study of the world glaciers by coupling the simple avalanche model SnowSlide to the global glacier mass balance model Open Global Glacier Model (OGGM). The evolution of this contribution throughout the 21st century is presented, using different emission scenarios. The authors quantify the global avalanche contribution to glacier accumulation (3%) and erosion (1%), with stronger contributions in some regions like New Zealand and for smaller glaciers. The spatial variability of glacier mass balance is increased by avalanches, with local impacts on individual glacier evolutions. At regional scale, avalanches have little impact on the evolution of glacier volumes in the 21st century, but have a significant impact for small glacier with longer subsistence. The avalanche contribution itself has varying evolutions in the 21st century depending on regions.

This study brings novel and useful insights, by explicitly adding the contribution of snow avalanches to glacier mass balance modelling. This explicit modelling improves existing empirical parameterizations which can't account for the non-linearity of this process. The worldwide application of the method throughout the 21st century provides very interesting results for the cryosphere community. The methods used are generally sound, properly justified and clearly presented. The paper is well written and clearly exposes its main conclusions. I only have a few minor comments and questions regarding the methods in

snow redistribution, detailed hereafter. Note that my review may be incomplete since my expertise does not cover glacier modelling itself.

Specific minor comments

- Parameterization of SnowSlide

The calibration of the snow holding depth following the slope differs from the original one. Why? Can you indicate how this function was calibrated, e.g. where and using what data? Do you think the calibration could be different depending on the regions, their topography and their snow climatology?

- Influence of varying spatial resolution

As you mentioned, smaller grid spacing on smaller glaciers leads to more avalanching than on larger glaciers. The study of sensitivity to spatial resolution is useful and interesting. However, it could be supplemented by showing sensitivity for the most extreme cases (the smallest glaciers), since these are an important focus of the main results. How would the results change for a very small glacier (e.g. Moore Glacier in Fig. 3) where spatial resolution is around 20 m, if this resolution was the same as for the largest glaciers (200 m)? Would the outcomes for these very small glaciers change significantly?

Technical comments and typos

- The readability of some figures, particularly Fig. 2, can be improved: the text is sometimes hard to read at 100% zoom.

- The following comment is more of a note, as I assume it is of limited significance in the context of global modelling and long-term climate projections.

Aspect dependency: I understand that no snowpack model is used prior to SnowSlide, which is, of course, justified in the context of the study. But since the snow depth used by SnowSlide is a proxy for snowfall, doesn't it lead to avalanching in sun-exposed faces being relatively overestimated versus shaded faces, where less snow melt and compaction could occur before avalanching?

- L. 34: warming

- L. 64-65: I assume you mean: snow redistribution by wind

- L. 129: "absolute" or relative?

- L. 272: Fig. 7-24

(Remarks on code availability)

Reviewer #3

(Remarks to the Author)

I read this model study with great interest and was impressed by the authors' efforts to quantify the avalanche contribution for glaciers worldwide. This work represents a valuable contribution, and the dataset provided offers an excellent basis for further analyses of the factors driving avalanche contributions to glaciers and how these may evolve under a changing climate. This study constitutes a very interesting first step. However, I have several comments and concerns that I believe should be addressed before publication.

Main comments:

Avalanche process representation: Avalanche contributions to glaciers are often linked to enhanced snow accumulation resulting from preferential deposition on surrounding slopes. As far as I understand, this process is not explicitly accounted for in the study. Instead, the avalanche contribution appears to be modeled primarily as a function of topography (slope angle and elevation of adjacent terrain and the glacier terrain), which may not fully capture the underlying processes. This simplification could lead to an underestimation of avalanche effects, particularly for many small glaciers. That said, I recognize the necessity of using a simplified approach when modeling avalanche contributions at a global scale, and the study represents an important step forward. However, I would encourage the authors to include a critical discussion of this limitation and its potential impact on the model results.

Model time step: Coming from a background in small-scale studies, I wonder why the authors chose a relatively coarse model time step of one month. How does this affect the representation of precipitation distribution and the partitioning between solid and liquid precipitation?

Section order: I was surprised by the ordering of the sections. I would have preferred reading the methods before the results, as this structure provides the necessary foundation for understanding what the authors have actually done.

Minor comments:

Introduction: Please provide an overview of the range of avalanche contributions estimated in previous studies for individual glaciers.

Results – topographic relationships: It would be valuable to analyze the relationship between avalanche contribution (in percentage) and the aspect, slope, and elevation difference of surrounding slopes and the glacier slopes that contribute to the overall mass to glaciers. This could help assess which glaciers may benefit more from avalanches in the future, separated by these topographic characteristics.

Clarity in results and discussion: I appreciate the analysis of potential future effects of avalanches on glacier mass balance. However, the presentation of the results and discussion is currently quite wordy and sometimes confusing. I recommend rewriting this section to present the information more clearly, emphasizing the interdependencies between processes, climate change, and glacier size evolution.

Precipitation processes: Please discuss the effects of missing precipitation processes and their strong spatial variability, particularly in high-elevation and steep mountain terrain. Include how these omissions could influence your projections for the future.

Methods section:

Variable grid size: The use of a variable grid size depending on glacier size is understandable, but I suggest discussing the potential effects on model results. Changing the grid size affects the representation of topography, slope angles, and therefore snow redistribution, which impact the model outputs and introduces a bias for smaller versus larger glaciers.

Minimum slope angle of 25°: Please provide reasons for not using the minimum slope angle of 25°.

Figures:

- The fonts in many of the figures are very small and difficult to read; please increase the font size for better clarity.

Additionally, the quality of Figure 1 appears somewhat low and could be improved.

- Figure 1: I do not clearly understand the distinction between “1. Snow removal” and “4. Snow redistribution.” In the figure, “Snow removal” also seems to indicate redistribution from very steep, high-elevation slopes to flatter, low-elevation areas of the glacier. Please revise this figure to make the processes and their differences more clear.

(Remarks on code availability)

Version 1:

Reviewer comments:

Reviewer #1

(Remarks to the Author)

Dear Marin,

Thank-you for your considered responses to all the reviewers feedback on your manuscript. The detail provided and edits made really strengthen this contribution. This is a great piece of work, which I am sure will generate much interest. I look forward to seeing this work published.

Kind regards

Heather

(Remarks on code availability)

There appears to be good documentation and links to other workbook to help others utilise this code. Although I am not a modeller I am (with the help of a visiting academic) currently teaching the OGGM model to a post-graduate class. My aspiration is to utilise the OGGM in future for teaching and research. I greatly appreciate that the authors sharing the code and making it available to others for future work.

Reviewer #2

(Remarks to the Author)

The authors provided thorough answers to the reviewers in general, and to my comments in particular. The additions and modifications improve the quality of the manuscript. I recommend publication. Congratulations for this very interesting study.

(Remarks on code availability)

Reviewer #3

(Remarks to the Author)

All of my questions and comments have been successfully addressed, and the manuscript has been thoroughly and appropriately revised. I am satisfied with the changes made, and I recommend the manuscript for publication.

(Remarks on code availability)

REVIEWER COMMENTS

Reviewer #1 (Remarks to the Author):

This is a valuable contribution that explores the role of avalanche to glacier mass gain (or loss) at the global scale – the first time such an analysis has been applied to glaciers globally. The authors couple an open source (OGGM) glacier model to a gravitational snow redistribution model (Snow Slide) to explore glacier mass balance with and without modelled avalanche input. Model parameters were calibrated to match the geodetic mass balance developed by Hugonnet et al (2021) and results are also compared to glaciological mass balance data. Future projections were then made for using three SSPs exploring how avalanche accumulation may affect glacier evolution in a warming world. This is a high-quality contribution, it is well written, well cited, and contains a number of quality Figures to support findings. I enjoyed reading this contribution and have only minor comments below.

We'd like to thank reviewer 1 for their comments and thorough review, this is really appreciated.

Specific comments to each section follow:

Title: Perhaps the term 'variable' rather than 'significant' would be better in the title? While the percentage contribution of avalanche input is indeed significant in some glacier regions, many other regions had contributions of less than 1%. When considering Supplementary Fig 6, one might determine that across the global sample the avalanche contribution was not 'significant' but quite significant particularly in alpine settings.

We've changed the title to *'Topographically-controlled contribution of avalanches to glacier mass balance in the 21st century'*.

Line 34: warming - minor typo 'warmin'

Good catch. Modified (L36).

Fig 1: This is a nice Figure, but the blue lines in 1a are a bit confusing. Does the blue line indicate a hypothetical glacier area?

We have modified this figure to distinguish the avalanche release, transition and deposit areas and added a legend for the glacier outlines and mountain ridges. We have also changed the color of the glacier outlines & crevasses so that it is not confused with the surface mass balance 'Control' line:

Figure 1: Impact of avalanches occurring at the surface of a hypothetical mountain glacier. a Hypothetical glacier with avalanche release, transition and deposit areas in light blue, purple and red, respectively. **b** Corresponding effect on the annual mass balance patterns with elevation. Avalanches occurring within the glacier boundaries (1-3 & 6) lead to additional accumulation on the avalanche deposits (3) while the removal of snow in the release areas locally reduces the surface mass balance (1). Avalanches from outside (4-5) the glacier boundaries, from seasonally snow-covered slopes or other glaciers, lead to a local increase in mass balance.

Line 66: Reference to the ‘temperature bias’ can this be explained more clearly. Is this referring to a known bias associated with the gridded temperature data, to the known variability of temperature with elevation, or, be a reference to previous reports of elevation dependant warming? A few additional sentences to explain exactly what temperature bias authors are referring would be appreciated.

This ‘temperature bias’ is a parameter of the mass balance model used in OGGM - an additive parameter to the temperature supposed to correct potential offsets in the climate reanalysis temperature series, indeed for the reasons the reviewer mentions (the bias parameter is temporally constant however, i.e. it cannot account for elevation dependant warming). We’ve now clearly specified that this is a model parameter (L70-72):

‘This precipitation correction factor, together with a temperature bias parameter and the melt factor used in global-scale temperature-index models, needs to be calibrated on a glacier-by-glacier basis²⁸.’

We note that this is explained in detail in the Methods (Recalibration of the glacier model). We therefore do not insist on this here in the Introduction.

Line 91: Model simulations have been set-up to align with calendar years. Was there any consideration to using the hydrological year – even though northern and southern hemisphere will be opposing? Starting simulations in January means that the first ablation (accumulation) season is half over in the southern (northern) hemisphere respectively, although equatorial glaciers will be different again. Perhaps this is why the start of the time series look a bit odd in Fig 6 for the NZ region? But maybe this does not matter depending on model spin-up processes?

Thanks for this thoughtful input. The period 01/2000-12/2019 had to be the same as the glacier model calibration period, which directly has to be identical to the geodetic mass balance estimates from Hugonnet et al. (2021), as is done for other global modeling studies (Rouze et al., 2023; Zekollari et al., 2025). This was necessary to stay consistent between the period over which the avalanche inputs were calculated, and the glacier model calibration period. As a result, that indeed means that we start in January both with the glacier and avalanche simulations but we would expect this effect to be minor and just affect the very first few months - note that the avalanche contribution was then averaged over this whole period, so these first few months are unlikely to have any influence on the model runs.

New Zealand does show an increase in glacier area at the start of the historical period (which we guess is what was meant when pointing at the oddness in Fig. 6), contrary to the other regions (Fig. 6; Fig. S25-S40). This could be due to an initial shock due to the absence of model spinup, but this is unlikely as this is not visible for the other regions. Perhaps this could be related to the small glacier advance that occurred in the early 2000s due to some cooler temperatures at the time (as suggested by Mackintosh et al., 2017). This also is visible in both satellite remote sensing (Zemp et al., 2025) and in situ mass balance observations (Sirguey et al., 2016; Cullen et al., 2019).

We have now specified in the text that 01/2000-01/2019 is the model calibration period to clarify this choice (L95-97):

'We use a simple gravitational snow redistribution model^{31,32} run with W5E5v2.0 mean monthly climate inputs³³ over the glacier model calibration period 01/2000-12/2019 to quantify the contribution of avalanches to the annual glacier accumulation (Methods).'

Line 108 (plus Sup): It is acknowledged that the Randolph Glacier Inventory is the 'standard' for global glacier outlines. However, there exists some rather inaccurate outlines for some glaciers. For example, the RGI outline of Rolleston Glacier (NZ) is at best a poorly aligned representation of the glacier back in the 1980-90s. It is significantly oversize (and misaligned). Since at least 2010 the entire portion of the RGI outline to the south-east is no longer connected to the current glacier, it is well separated by bedrock and not considered part of the glacier. The slopes above this now non-existent part of the glacier are prone to avalanche deposition, so one might expect this will create a positive bias in results. Without in-depth knowledge of all glacier regions, one wonders how many other small glaciers suffer the same poor alignment of RGI outlines, which will consequently impact the accuracy of the geodetic mass balance as produced by Hugonnet et al and results of this study. Did the authors undertake spot-checks of the glacier areas? I appreciate that this is perhaps beyond the scope of the analysis, but such errors are potentially perpetuating from one global study to the next.

We fully acknowledge the limitations of RGI 6.0, which we depended on in this study given that the geodetic mass balance estimates needed for the calibration were produced specifically for this inventory (Hugonnet et al., 2021). Rolleston is a typical example of these limitations and this was one of the reasons that we did not attempt to validate our model at this site, despite it being one of the rare glaciers with estimates of avalanche contribution (Purdie et al., 2015). As a result, the glacier-scale simulations need to be interpreted with caution and we showed them only to indicate the variability in avalanche influence between different glaciers (Fig. 3). Currently all global-scale glacier modeling studies rely on the RGI (until now RGI 6.0, given that the main glacier products - ice thickness, geodetic mass balance - have not yet been adapted to RGI 7.0).

We did not conduct spot-checks ourselves as this was indeed far beyond the scope of this global-scale study, and such limitations from the RGI 6.0 and their influence on model results have been described already in the scientific literature (e.g. Hartl et al., 2025). However, we refer the reviewer to the work by Aguayo et al. (2024) who assessed the uncertainty of OGGM in the Chilean Andes resulting from the choice of glacier inventory. They compared the future glacier projections obtained with RGI 6.0 and the with the Chilean national inventory, but also with various ice thickness estimates or climate scenarios and demonstrated that ‘While the glacier inventory and ice thickness source showed overall differences close to 10 %, the different climate alternatives showed differences of more than 50 % for solid precipitation, for example.’ (Aguayo et al., 2024).

Ultimately, we note that this study aims to provide a first estimate of the influence of avalanches on glacier mass balance at the global scale. It is only a first step that will undoubtedly be far from perfect but has the advantage of putting some numbers on a process that was until now completely ignored.

We however agree that these limitations need to be clearly stated and discussed and have now expanded their description in the Methods (L488-492):

‘We note that the use of RGI 6.0 outlines, for example due to potential misalignments, can lead to uncertainties in glacier future projections and also most likely in estimates of the avalanche contribution⁶¹. At the regional scale the uncertainties are small relative to other contributions such as from the choice of climate models⁶², but for individual glaciers these estimates should be interpreted with caution.’

Lines 136-138: The results relating to where on the glacier avalanche deposition occurs is very interesting and ideally warrants further investigation/discussion. Avalanche deposition onto the ablation zone creates a variable range of feedbacks, especially in relation to albedo/melting. Avalanche deposition onto an ablation area may increase or decrease albedo depending on surface type (ice/debris) and avalanche deposition is known to be of higher density, thus more mass. Could the authors say a little more about the differences between glaciers where avalanche deposition is primarily occurring in the accumulation area of glacier compared to avalanche deposition the occurs in glacier ablation areas?

Thank you. Please see our answer to the comment below for the comparison of glaciers with avalanche contribution in the ablation versus accumulation area.

For the feedbacks related to albedo and density, this is an excellent point, which however cannot be addressed with the simple model that we used: the OGGM mass balance model that we use does not include any representation of snow metamorphism, nor therefore albedo or density - see Methods. This choice was made to be able to get a first global estimate of the avalanche contribution and its primary effect on glacier evolution, but considerably simplifies the related processes. Almost all global glacier models tend to rely on models with a limited number of parameters to reduce the equifinality when calibrating against the geodetic mass balance only (Schuster et al., 2023), and we therefore stayed with this choice to be able to compare the runs with avalanches to the OGGM standard simulations.

We are currently working on a field-based study where we aim to disentangle these different processes occurring on one avalanche deposit of Argentière Glacier (French Alps), but this is a whole different study conducted at a completely different spatial scale. For this paper, we have however now acknowledged these limitations in the main text (L407-414):

'A major challenge, however, lies in calibrating and validating gravitational snow redistribution models in glacierized areas due to the scarcity of remote sensing and field-based observations. Expanding such datasets will be crucial to refine future model estimates (Methods), particularly to capture feedbacks related to avalanche deposition, such as effects on albedo and snow density, and to better understand how avalanche activity depends on the characteristics of release areas (aspect, slope, or hanging glaciers).'

Line 168: It would seem that a number of regions required almost no recalibration of the precipitation factor, which would indicate that at these sites the role of avalanche input (either positive or negative) is negligible. Results shown in Sup Fig 5 which are close to null, align with results in Fig 2 for areas where avalanche processes are negligible. One might conclude that in general it is glaciers in mountainous terrain for which avalanche deposition is important. Perhaps this finding could be stated more explicitly.

Absolutely. We have insisted on this in the main text (L175-178):

'Given that avalanches tend to add mass to glaciers at the regional scale (Fig. 2), accounting for avalanches generally leads to lower precipitation correction values, with mean regional differences up to 0.1 for Central Europe (Supplementary Figure 5). This is especially the case in regions with steep relief, which are more prone to avalanching (Fig. 2).'

Line 175-200: The individual glacier case studies are interesting and a great way to tease out specific examples. Could the authors include a Figure (possibly in supplementary) that supports Fig 3 so readers can quickly ascertain exactly where in the regional mountain ranges the various case studies are located.

We have added a Supplementary Figure 6 showing the precise location of these glaciers:

Supplementary Figure 6: Location of the four focus glaciers. The glaciers are indicated by a red triangle and the blue outlines in the background show the RGI 6.0 glacier outlines in the corresponding regions: **a** Argentière Glacier (RGI6.0-11.03638; 45.951°N, 6.985°E), **b** Moore Glacier (RGI6.0-18.02066; -43.898°N, 169.786°E), **c** Black Glacier (RGI6.0-01.13640; 61.085°N, -143.831°E) and **d** Bazhin Glacier (RGI6.0-14.20029; 35.226°N, 74.637°E).

Line 305-307: As glaciers thin slope angle tends to increase in the headwall region this may be an additional ‘future’ feedback.

Absolutely, this is partly what we implied when mentioning ‘glacier surface lowering’. We have specified this in more details in the Methods (L575-579):

‘This suggests that although glacier surface lowering (creating steeper and taller headwalls) and a rising rain-snow transition may affect avalanche-driven mass redistribution in the future^{20,49}, these influences are likely secondary compared to the main climatic controls on glacier evolution.’

Line 311-315: The Southern Alps of NZ are relatively low elevation compared to mountain ranges around the globe, perhaps this lack of 'real-estate' at higher elevation contributes to the finding that avalanche contribution strongly depends on the SSP. As temperatures warm the snow/rain threshold will likely be off the top of many mountains in NZ.

This is an excellent suggestion, that we have added to the main text (L329-334):

'In New Zealand, the avalanche contribution depends strongly on the SSP scenario: it is expected to remain at its current level under SSP1-2.6, but decrease in higher emission scenarios, from 15% in 2020 to 3% in 2100 under SSP5-8.5 as the glacierized area shrinks (Fig. 6d-f). This likely reflects a dependence on the rising snow-line which, in high-warming scenarios, would rise above most of the relatively low summits in this range.'

Lines 350-364: The authors have highlighted some interesting results; this is a great study. However, I think that considering results only in terms of 'geopolitical' regions limits the findings. What are the patterns for cirque glaciers globally compared to valley glaciers (regardless of region)? What about debris-covered glaciers? It appears that the Himalayan glaciers create a bias, I can think of many debris-covered glacier for which avalanche accumulation is not an important contribution. How do glaciers that have significant avalanche deposition in their ablation areas compare to those where avalanche is confirmed to the upper accumulation areas? These are all very interesting spatial questions for which the authors have results, but which are not discussed?

We really appreciate these inputs, that refer to the potential of this new dataset to explore relationships and test the influence of avalanches for various types of glaciers. There would be a lot to do, but at this stage we mainly wanted to provide a first quantification geared towards regional values, along with open-access to the code and dataset, to allow for more in-depth analysis at various spatial scales. We therefore refrained from adding too much of these analyses to the text. The different points that were raised are however definitely worth exploring and we provide short answers to these questions below. This as led us to make a few modifications of the text that we also list here:

- **Debris-covered glaciers:** many regions show a significant positive correlation between debris cover and avalanching (Fig. S2-S4), but there is a lot of spread so this obviously doesn't explain the complete signal. This is true not just for the HMA regions, but also for Alaska, Western Canada and USA, Central Europe or New Zealand, especially when considering only the glaciers receiving a net accumulation surplus by avalanching (Fig. S3). We totally agree that many debris-covered glaciers are not avalanche-fed and vice versa. We show however that there is a certain tendency of debris-covered glaciers to also be avalanche-fed (which does not indicate causality!). These correlations actually likely come from the fact that debris-covered glaciers and avalanche-fed glaciers alike need to be surrounded by steep terrain for debris OR snow to be deposited on their surface. But we are indeed looking at two different kinds of processes here. We have highlighted this in the description of these correlations (L141-143):

'This confirms past observations conducted at the scale of High Mountain Asia⁴¹, which likely indicate that debris-covered and avalanche-fed glaciers exist in a similar topographic setting, with steep surrounding slopes.'

- **Cirque versus valley glaciers:** This would be an excellent comparison, albeit a not so straightforward one at the global scale, which was the focus of this study. Indeed, the classification does not exist for all 200,000+ RGI glaciers. It would be possible only for a subset of glaciers for which these attributes have been indicated in the GLIMS database. As to whether these glaciers would be influenced differently by avalanches definitely calls for more analysis which is however beyond the scope of this study.
- **Avalanches affecting the ablation versus accumulation areas:** yes, this is also a good point, which could potentially explain the different behaviours of Black and Bazhin Glaciers despite their similar avalanche contribution, in Figure 3. This is also what we hinted at in the main text (L211-216):

'These differences in glacier projections for a similar glacier-wide avalanche contribution come from the local topography which can result in some cases in snow being redistributed from the accumulation zone into the ablation zone, thereby enhancing mass loss, and in other cases in the expansion of the accumulation zone by increasing the mass supply from higher elevation (Fig. 1).'

However looking at this at the large scale is far from trivial for three main reasons: 1/ many glaciers receive avalanches both in their ablation and accumulation areas, some also lose snow by avalanching from both. A classification - while definitely possible - is thus not straightforward and would require defining a specific metric to characterize this ratio. 2/ The accumulation-ablation area ratio is changing with time, sometimes rapidly with climate change, and different mountain regions are responding more or less fast to this climate change. This makes it challenging to define a reference time at which to make this comparison. 3/ Accounting for avalanches leads to a change of definition of 'accumulation' and 'ablation' zone. Removing snow from the accumulation area may lead to parts of this area to have a negative mass balance while an avalanche deposit in what we traditionally categorize as 'ablation' area could have positive mass balance (Fig. 3).

Despite the underlying challenges, we do believe that such analysis would be worthwhile and should be undertaken, but we felt that this was beyond the scope of this study. We have however now acknowledged this potential for new analysis in the main text (L388-392):

'This study also paves the way for more detailed sub-regional studies, for example examining links between avalanches and debris cover (Supplementary Figures 2-3), differences in avalanche effects across glacier types (valley, cirque), or the impacts of avalanches occurring in accumulation versus ablation zones (Fig. 3).'

Methods section: This was well written and clear to follow. I found that questions I had on reading the core paper e.g. impact of DEM resolution, role of precipitation factors etc were all answered in due course in this section.

Thank you.

Supplementary: The supplementary is substantial, but all the material is well presented and strongly supports the main document. It was good to see all the regional graphs, although seeing some

Figures/results focused on glacier characteristics rather than glacier regions would have been a great addition.

Thanks, it's actually really exciting to explore this new dataset and look at individual glaciers. However, as explained in the comments above, we decided to stick to the regional estimates, which are also less uncertain. Figure 3 was the exception to show a few examples of where the regional patterns could come from. This paper will be complemented by a more process-focused study to be submitted in the coming months.

Reviewer #1 (Remarks on code availability):

I fully support the authors providing the code and this is likely something I will visit and explore in due course, and potentially utilise for post-graduate projects. However, I have not (as yet) used the OGGM, so did not check the code.

Thanks, we'll be very happy if this code can be used by others! The documentation is not (yet) to the standard of OGGM's but hopefully it is not too difficult to handle.

Reviewer #2 (Remarks to the Author):

Review of: "Substantial contribution of avalanches to glacier accumulation in the 21st century", M. Kneib et al.

General comment

This paper explores the contribution of gravitational snow redistribution to glacier accumulation, through an extensive study of the world glaciers by coupling the simple avalanche model SnowSlide to the global glacier mass balance model Open Global Glacier Model (OGGM). The evolution of this contribution throughout the 21st century is presented, using different emission scenarios. The authors quantify the global avalanche contribution to glacier accumulation (3%) and erosion (1%), with stronger contributions in some regions like New Zealand and for smaller glaciers. The spatial variability of glacier mass balance is increased by avalanches, with local impacts on individual glacier evolutions. At regional scale, avalanches have little impact on the evolution of glacier volumes in the 21st century, but have a significant impact for small glacier with longer subsistence. The avalanche contribution itself has varying evolutions in the 21st century depending on regions.

This study brings novel and useful insights, by explicitly adding the contribution of snow avalanches to glacier mass balance modelling. This explicit modelling improves existing empirical parameterizations which can't account for the non-linearity of this process. The worldwide application of the method throughout the 21st century provides very interesting results for the cryosphere community. The methods used are generally sound, properly justified and clearly presented. The paper is well written and clearly exposes its main conclusions. I only have a few minor comments and questions regarding the methods in snow redistribution, detailed hereafter. Note that my review may be incomplete since my expertise does not cover glacier modelling itself.

We'd like to thank reviewer 2 for their comments and thorough review, this is really appreciated.

Specific minor comments

- Parameterization of SnowSlide

The calibration of the snow holding depth following the slope differs from the original one. Why? Can you indicate how this function was calibrated, e.g. where and using what data? Do you think the calibration could be different depending on the regions, their topography and their snow climatology?

Thanks for this comment. Indeed, the parameters we used (0.14; 145) vary slightly from the original ones (0.12; 120) - or at least these are the values that we found when digitizing the Fig. 2b plot from Bernhardt & Schulz (2010), originally obtained based on the comparison between end-of-season snow extents from Landsat ETM+ satellite images over the Watzmann massif.

First, we note that the differences between the maximum snow depth as a function of slope for the different parameter pairs ([0.14; 145] VS [0.12; 120]) are very small, as indicated in the figure below (Fig. R1), which translates to the avalanche deposit extents and volumes being also similar for both parametrizations.

Figure R1: Maximum snow depth as a function of slope relationships using two different sets of parameters for the snow redistribution model.

The parametrization that we use tends to remove a bit more snow from slopes in the [10; 40]° range than the original parametrization. This parametrization was obtained based on a qualitative assessment of SnowSlide outputs in the Mt Blanc massif, and Argentière Glacier specifically, where we could directly compare the OGGM-SnowSlide outputs with distributed mass balance data from an ice flux inversion ²¹. There, we found a good agreement with the runs at monthly time scale and the additional parametrization of inertia from Quéno et al. (2024). In our study we further evaluated the avalanche extents obtained with this parametrization against the Sentinel-1 avalanche deposit extents from Kneib et al. (2024a), as described in the methods.

We have now mentioned this in the methods (L458-466):

'For this study, we use values of 0.14 and 145 for the parameters a and C , respectively, along with the same parameterization of avalanche inertia introduced by ³². We did not use a minimum slope threshold as we did not see any difference when including it. The values of a and C differ slightly from those originally reported by ³¹ – 0.12 and 120, respectively. They are obtained based on a qualitative assessment of SnowSlide outputs in the Mt Blanc massif, and Argentière Glacier specifically, where we could directly compare our model outputs with distributed mass balance data from an ice flux inversion ²¹. This parametrization was further evaluated against Sentinel-1 avalanche deposits (Methods - Avalanche model evaluation).'

Based on this, we could also envision that the SnowSlide parameters may need to be recalibrated on a regional basis, most likely more because of the snow climatology than the topography. This would be a very interesting test, but which would require the appropriate data for these calibration runs, ideally along the lines of what Quéno et al. (2024) did using repeat snow depth measurements over large areas using data from an airborne LiDAR. This is now mentioned in the discussion (L407-414):

'A major challenge, however, lies in calibrating and validating gravitational snow redistribution models in glacierized areas due to the scarcity of remote sensing and field-based observations. Expanding such datasets will be crucial to refine future model estimates (Methods), particularly to capture feedbacks related to avalanche deposition, such as effects on albedo and snow density, and to better understand how avalanche activity depends on the characteristics of release areas (aspect, slope, or hanging glaciers).'

- Influence of varying spatial resolution

As you mentioned, smaller grid spacing on smaller glaciers leads to more avalanching than on larger glaciers. The study of sensitivity to spatial resolution is useful and interesting. However, it could be supplemented by showing sensitivity for the most extreme cases (the smallest glaciers), since these are an important focus of the main results. How would the results change for a very small glacier (e.g. Moore Glacier in Fig. 3) where spatial resolution is around 20 m, if this resolution was the same as for the largest glaciers (200 m)? Would the outcomes for these very small glaciers change significantly?

Thanks for this comment. Actually Supplementary Figures 44 & 45 d-f show the effect of lowering the spatial resolution to 50 and 100 m for all the Central Europe glaciers smaller than 1 km² (so with a resolution of 24 m or higher). These results show significant changes especially when going from 50 m

to 100 m resolution, less when going from the original resolution to 50 m (Fig. 5), but the overall patterns are maintained.

We get similar results when looking at this on a glacier-by-glacier basis, as shown below for two small glaciers in Central Europe, where the altitudinal mass balance and future volume changes are particularly affected when going from 50 m to 100 m resolution, less so in the step from the original resolution to a resolution of 50 m. We have added this as two new figures in the supplementary information to complement the regional-scale observations, and added a sentence on this topic to the main text:

Main text (L639-646):

'Here, we test the sensitivity of the avalanche contribution and glacier volume changes to the spatial resolution of the DEMs by running our model at 50 m and 100 m resolution for Central Europe (Supplementary Table 3; Supplementary Figures 44-46). We find the regional avalanche contribution to be similar in all three cases, both for the historical period and for the future simulations, with the results from the reference simulations being more similar to those from the 50 m resolution simulations (Supplementary Figures 44 & 46). Such findings are also true at the scale of small individual glaciers (Supplementary Figures 47-48).'

Supplementary figures:

Supplementary Figure 47: Influence of DEM spatial resolution on volume change of Tré-les-Eaux Glacier. *a* Tré-les-Eaux Glacier in Central Europe with altitudinal mass balance and 21st century volume changes *b-c* at the original 20 m resolution, *d-e* at 50 m resolution and *f-g* at 100 m resolution. The left panel shows a 3D view of the glacier with the average avalanche correction factor (k_{ava}) over the period 01/2000-12/2019 indicated by the red-white-blue shade wrapped over the glacier extents from the Randolph Glacier Inventory (RGI) 6.0³⁹. The background topography is from the DEM used to run the OGGM simulations. The numbers indicate the glacier-wide avalanche contribution and the glacier area. The central panels show the altitudinal mass balance for the ‘Control’ and ‘Avalanches’ simulations, with the red dashed line corresponding to a mass balance value of 0 m w.eq. yr⁻¹. The right panels show the past and future volume changes of these glaciers as modelled using OGGM. All percentages are given relative to the initial glacier volume in 2000 in the Control scenario. The black line corresponds to the historical period over which the mass balance model was calibrated using W5E5v2.0 data³³. The colored lines show the median future projections for different SSP scenarios, and the shaded areas indicate the 25th-75th percentile range. The different curves were smoothed using a 5-year rolling mean.

Supplementary Figure 48: Influence of DEM spatial resolution on volume change of Arête Midi-Plan.

a Arête Midi-Plan in Central Europe with altitudinal mass balance and 21st century volume changes b-c at the original 20 m resolution, d-e at 50 m resolution and f-g at 100 m resolution. The left panel shows a 3D view of the glacier with the average avalanche correction factor (k_{ava}) over the period 01/2000-12/2019 indicated by the red-white-blue shade wrapped over the glacier extents from the Randolph Glacier Inventory (RGI) 6.0³⁹. The background topography is from the DEM used to run the OGGM simulations. The numbers indicate the glacier-wide avalanche contribution and the glacier area. The central panels show the altitudinal mass balance for the 'Control' and 'Avalanches' simulations, with the red dashed line corresponding to a mass balance value of 0 m w.eq. yr⁻¹. The right panels show the past and future volume changes of these glaciers as modelled using OGGM. All percentages are given relative to the initial glacier volume in 2000 in the Control scenario. The black line corresponds to the historical period over which the mass balance model was calibrated using W5E5v2.0 data³³. The colored lines show the median future projections for different SSP scenarios, and

the shaded areas indicate the 25th-75th percentile range. The different curves were smoothed using a 5-year rolling mean.

Technical comments and typos

- The readability of some figures, particularly Fig. 2, can be improved: the text is sometimes hard to read at 100% zoom.

We have now increased the font sizes of figures 5, 6 and figure 2:

We have also updated figures 1 & 3 for readability, increasing the font sizes and adding a divergent colorbar to figure 3:

- The following comment is more of a note, as I assume it is of limited significance in the context of global modelling and long-term climate projections.

Aspect dependency: I understand that no snowpack model is used prior to SnowSlide, which is, of course, justified in the context of the study. But since the snow depth used by SnowSlide is a proxy for snowfall, doesn't it lead to avalanching in sun-exposed faces being relatively overestimated versus shaded faces, where less snow melt and compaction could occur before avalanching?

This is an excellent point which we totally agree with. As the reviewer states, there was not really any alternative to SnowSlide to make these calculations at the global scale but we agree that this would be an important point to consider at smaller spatial scales. We have now added a sentence related to this point in the last section of the main text (L407-414):

'A major challenge, however, lies in calibrating and validating gravitational snow redistribution models in glacierized areas due to the scarcity of remote sensing and field-based observations. Expanding such datasets will be crucial to refine future model estimates (Methods), particularly to capture feedbacks related to avalanche deposition, such as effects on albedo and snow density, and to better understand how avalanche activity depends on the characteristics of release areas (aspect, slope, or hanging glaciers).'

- L. 34: warming

Good catch. Modified (L36).

- L. 64-65: I assume you mean: snow redistribution by wind

Indeed! Added (L70).

- L. 129: "absolute" or relative?

Good catch, we've only left 'the avalanche contribution' as the definition speaks for itself (L136).

- L. 272: Fig. 7-24

Thanks, this has been adapted to the new figure indexing.

Reviewer #3 (Remarks to the Author):

I read this model study with great interest and was impressed by the authors' efforts to quantify the avalanche contribution for glaciers worldwide. This work represents a valuable contribution, and the dataset provided offers an excellent basis for further analyses of the factors driving avalanche contributions to glaciers and how these may evolve under a changing climate. This study constitutes a very interesting first step. However, I have several comments and concerns that I believe should be addressed before publication.

Many thanks for this thorough review and comments which we've tried to address as best as we could.

Main comments:

Avalanche process representation: Avalanche contributions to glaciers are often linked to enhanced snow accumulation resulting from preferential deposition on surrounding slopes. As far as I understand, this process is not explicitly accounted for in the study. Instead, the avalanche contribution appears to be modeled primarily as a function of topography (slope angle and elevation of adjacent terrain and the glacier terrain), which may not fully capture the underlying processes. This simplification could lead to an underestimation of avalanche effects, particularly for many small glaciers. That said, I recognize the necessity of using a simplified approach when modeling avalanche contributions at a global scale, and the study represents an important step forward. However, I would encourage the authors to include a critical discussion of this limitation and its potential impact on the model results.

Thanks for this comment. Indeed our approach relies solely on precipitation and topography to estimate the avalanche contribution of each individual glacier. We do not account for any wind redistribution, which could locally enhance or reduce the effective snow depth, and therefore affect the avalanche supply. Wind speed and direction being a very localized process, it is challenging to take into account beyond the glacier/catchment scale, and such a modelling endeavour would likely be very difficult to calibrate/validate (Voordendag et al., 2024).

This was suggested in the original manuscript (L103-107):

'We note that this model does not represent wind-driven snow redistribution, which is a particularly difficult process to constrain given its dependence on an accurate representation of wind fields on a glacier, and despite recent advances in quantifying and modelling this effect at relatively small spatial scales^{32,36-38}.

We have also now specifically mentioned this point in the discussion (L420-432):

'Our results underscore the need for an improved representation of local processes affecting the spatial variability of mass balance in glacier models, which, beyond avalanches, include frontal ablation of marine- and lake-terminating glaciers, variable melt under debris, variable precipitation with elevation or snow redistribution by wind. Further research is needed to understand how these processes interact^{30,32,38,53} and how they can be effectively integrated into global glacier models^{6,8,30}. For example, wind-driven snow redistribution may locally enhance or reduce the effective snow depth on slopes on or in the vicinity of glaciers, and therefore affect the avalanche supply. These process representations, combined with the recent developments in remote sensing data availability and assimilation by global glacier models and the anticipated capability of running these simulations in a distributed way, will pave the way for a new generation of models capable of accurately projecting the future of every individual glacier across Earth's mountain ranges.'

Model time step: Coming from a background in small-scale studies, I wonder why the authors chose a relatively coarse model time step of one month. How does this affect the representation of precipitation distribution and the partitioning between solid and liquid precipitation?

This is a very good point. This choice was driven mainly by computational constraints and also because 1 month is the time-step at which OGGM is run, which is also the temporal resolution of the vast majority of current global glacier models (Zekollari et al., 2025; Wimberly et al., 2025). Therefore, we also used the same precipitation partitioning scheme than OGGM, which is applied to monthly precipitation & temperature values.

Schuster et al. (2023) specifically tested running OGGM at daily resolution versus monthly and did not find a systematic effect on glacier volume changes, as both models were calibrated against the same data. In that study, more than the precipitation partitioning, the authors emphasized the resulting changes in monthly melt as being responsible for the increase in the spread of volume changes - see a full discussion of this effect in paragraph 4.1.2 of Schuster et al. (2023).

Based on these considerations, we decided to stay consistent with the reference OGGM runs and keep a one-month time step for our own simulations. A higher temporal resolution for SnowSlide would anyway not have been realistic at this spatial scale.

Section order: I was surprised by the ordering of the sections. I would have preferred reading the methods before the results, as this structure provides the necessary foundation for understanding what the authors have actually done.

This order of the sections is imposed by the Journal, and we can therefore not change it.

Minor comments:

Introduction: Please provide an overview of the range of avalanche contributions estimated in previous studies for individual glaciers.

We have now added this (L50-53):

'In the few locations where on-glacier avalanching has been estimated, it is one of the main sources of mass inputs, with estimated avalanche contributions ranging between 9% and 92% of the annual accumulation^{13,14,19–22}.

Results – topographic relationships: It would be valuable to analyze the relationship between avalanche contribution (in percentage) and the aspect, slope, and elevation difference of surrounding slopes and the glacier slopes that contribute to the overall mass to glaciers. This could help assess which glaciers may benefit more from avalanches in the future, separated by these topographic characteristics.

This comment hints towards attempting to derive an avalanche contribution from topographic characteristics only, and without any mass redistribution modeling. We consider this to be beyond the scope of this study and give a detailed summary of our reasons below.

While we agree that this is a worthwhile objective, we object to doing this with our results for the following reasons:

1. These are only model results, and not measurements or estimates of the actual avalanche contribution. While such a comparison would be extremely worthwhile for *measured* avalanche contributions, we expect that if we were to conduct it with our modelling results, it would be biased with the SnowSlide assumptions (homogeneous snow depth with aspect, density assumptions, spatial resolution used...)
2. Conducting such an analysis would be equivalent to trying to simplify the already quite simplistic avalanche estimation scheme. The SnowSlide scheme already accounts for slopes and precipitation, and can already be run relatively efficiently at the large scale as demonstrated by this study. Our recommendation would therefore be to directly use this redistribution scheme to estimate the avalanche contribution to glacier mass balance rather than relying on a proxy.
3. We expect the contributions from the different topographic factors to be difficult to disentangle, and that there will not be a single metric that can explain the diversity of avalanche contributions. This is based on a preliminary analysis of the avalanche contribution as a function of the R-index, which is the proportion of slopes steeper than 30° within a glacierized catchment. This metric has been used as a proxy for the potential of avalanche contribution in past studies, and proven to be one of the best predictors for the avalanche contribution (Hughes, 2008; Kneib et al., 2024a). We compared the SnowSlide avalanche contribution to this R index for 478 glaciers for which the R-index had been calculated in a recent study by Kneib et al. (2024a). We find that while a certain value of R-index is a necessary condition for a glacier to have a given avalanche contribution, it does not explain the whole variability in avalanche contributions (Fig. R1). These results are similar to those of Kneib et al. (2024a) conducted with mapped Sentinel-1 avalanche deposits in the Mt. Blanc, Everest and Hispar regions.

Figure R1: Avalanche contribution (from this study) as a function of R-index (from Kneib et al., 2024a) for 478 glaciers in the Mt Blanc, Hispar and Everest regions. Each circle represents one glacier and the circle size is scaled to the glacier size.

Based on this and the fact that such an analysis would require a substantial amount of work to be conducted at the global scale, we have not taken this any further.

Clarity in results and discussion: I appreciate the analysis of potential future effects of avalanches on glacier mass balance. However, the presentation of the results and discussion is currently quite

wordy and sometimes confusing. I recommend rewriting this section to present the information more clearly, emphasizing the interdependencies between processes, climate change, and glacier size evolution.

We have been through the entire Results & Discussion and simplified the text as much as we could. Please refer to the tracked-changes version of the manuscript to see all the minor changes that were made.

Precipitation processes: Please discuss the effects of missing precipitation processes and their strong spatial variability, particularly in high-elevation and steep mountain terrain. Include how these omissions could influence your projections for the future.

This is of course particularly relevant, not only for avalanches but for all the accumulation processes. Since all we did in this study was to add the representation of a process in the OGGM mass balance model, we do not expect the uncertainty of the precipitation to change between the model with and without explicit representation of avalanches. Regional glacier models correct precipitation based on glacier-specific calibration (Schuster et al., 2023), and this correction is a major source of uncertainty in regional simulations and is the main cause of inter-model runoff differences (Marzeion et al., 2020; Wimberly et al., 2025).

We have now specifically mentioned the lack of representation of variable precipitation (L420-432):

'Our results underscore the need for an improved representation of local processes affecting the spatial variability of mass balance in glacier models, which, beyond avalanches, include frontal ablation of marine- and lake-terminating glaciers, variable melt under debris, variable precipitation with elevation or snow redistribution by wind. Further research is needed to understand how these processes interact^{30,32,38,53} and how they can be effectively integrated into global glacier models^{6,8,30}. For example, wind-driven snow redistribution may locally enhance or reduce the effective snow depth on slopes on or in the vicinity of glaciers, and therefore affect the avalanche supply. These process representations, combined with the recent developments in remote sensing data availability and assimilation by global glacier models and the anticipated capability of running these simulations in a distributed way, will pave the way for a new generation of models capable of accurately projecting the future of every individual glacier across Earth's mountain ranges.'

Methods section:

Variable grid size: The use of a variable grid size depending on glacier size is understandable, but I suggest discussing the potential effects on model results. Changing the grid size affects the representation of topography, slope angles, and therefore snow redistribution, which impact the model outputs and introduces a bias for smaller versus larger glaciers.

Please see our detailed response to the comment from Reviewer 2.

Minimum slope angle of 25°: Please provide reasons for not using the minimum slope angle of 25°.

When testing the model in various mountain ranges we did not see any difference when including this minimum slope threshold, which is why we did not use this condition. We have now specified this in the text (L459-460):

'We did not use a minimum slope threshold as we did not see any difference when including it.'

Figures:

- The fonts in many of the figures are very small and difficult to read; please increase the font size for better clarity. Additionally, the quality of Figure 1 appears somewhat low and could be improved.

We have now increased the font size of figures 2, 3, 5 and 6. For figure 1, please see our detailed response to the comment below.

- Figure 1: I do not clearly understand the distinction between “1. Snow removal” and “4. Snow redistribution.” In the figure, “Snow removal” also seems to indicate redistribution from very steep, high-elevation slopes to flatter, low-elevation areas of the glacier. Please revise this figure to make the processes and their differences more clear.

We have modified this figure to distinguish the avalanche release, transition and deposit areas and added a legend for the glacier outlines and mountain ridges. We have removed the process descriptions “1. Snow removal” and “Snow redistribution” and used a more explicit terminology to describe the elements of the figure:

Figure 1: Impact of avalanches occurring at the surface of a hypothetical mountain glacier. a Hypothetical glacier with avalanche release, transition and deposit areas in light blue, purple and red. **b** Corresponding effect on the annual mass balance patterns with elevation. Avalanches occurring within the glacier boundaries (1-3 & 6) lead to additional accumulation on the avalanche deposits (3)

while the removal of snow in the release areas reduces the surface mass balance (1). Avalanches from outside (4-5) the glacier boundaries, from seasonally snow-covered slopes or other glaciers, lead to a local increase in mass balance.

REFERENCES

- Aguayo, R., Maussion, F., Schuster, L., Schaefer, M., Caro, A., Schmitt, P., Mackay, J., Ultee, L., Leon-Muñoz, J., & Aguayo, M. (2024). Unravelling the sources of uncertainty in glacier runoff projections in the Patagonian Andes (40–56° S). *The Cryosphere*, 18(11), 5383–5406. <https://doi.org/10.5194/tc-18-5383-2024>
- Bernhardt, M., & Schulz, K. (2010). SnowSlide: A simple routine for calculating gravitational snow transport. *Geophysical Research Letters*, 37(11). <https://doi.org/10.1029/2010GL043086>
- Cullen, N. J., Gibson, P. B., Mölg, T., Conway, J. P., Sirguey, P., & Kingston, D. G. (2019). The influence of weather systems in controlling mass balance in the Southern Alps of New Zealand. *Journal of Geophysical Research: Atmospheres*, 124, 4514–4529. <https://doi.org/10.1029/2018JD030052>
- Hartl, L., Schmitt, P., Schuster, L., Helfricht, K., Abermann, J., & Maussion, F. (2025). Recent observations and glacier modeling point towards near-complete glacier loss in western Austria (Ötztal and Stubai mountain range) if 1.5 °C is not met. *The Cryosphere*, 19(3), 1431–1452. <https://doi.org/10.5194/tc-19-1431-2025>
- Hughes, Philip. D. (2008). Response of a montenegro glacier to extreme summer heatwaves in 2003 and 2007. *Geografiska Annaler: Series A, Physical Geography*, 90(4), 259–267. <https://doi.org/10.1111/j.1468-0459.2008.00344.x>
- Hugonnet, R., McNabb, R., Berthier, E., Menounos, B., Nuth, C., Girod, L., Farinotti, D., Huss, M., Dussaillant, I., Brun, F., & Käab, A. (2021). Accelerated global glacier mass loss in the early twenty-first century. *Nature* 2021 592:7856, 592(7856), 726–731. <https://doi.org/10.1038/s41586-021-03436-z>
- Jouberton, A., Shaw, T. E., Miles, E., Kneib, M., Fugger, S., Buri, P., McCarthy, M., Kayumov, A., Navruzshoev, H., Halimov, A., Kabutov, K., Homidov, F., & Pellicciotti, F. (2025). Snowfall decrease in recent years undermines glacier health and meltwater resources in the Northwestern Pamirs. *Communications Earth & Environment*, 6(1), 691. <https://doi.org/10.1038/s43247-025-02611-8>
- Kneib, M., Dehecq, A., Brun, F., Karbou, F., Charrier, L., Leinss, S., Wagnon, P., & Maussion, F. (2024). Mapping and characterization of avalanches on mountain glaciers with Sentinel-1 satellite imagery. *The Cryosphere*, 18(6), 2809–2830. <https://doi.org/10.5194/tc-18-2809-2024>
- Mackintosh, A. N., Anderson, B. M., Lorrey, A. M., Renwick, J. A., Frei, P., & Dean, S. M. (2017). Regional cooling caused recent New Zealand glacier advances in a period of global warming. *Nature Communications*, 8(1), 14202. <https://doi.org/10.1038/ncomms14202>

Marzeion, B., Hock, R., Anderson, B., Bliss, A., Champollion, N., Fujita, K., Huss, M., Immerzeel, W. W., Kraaijenbrink, P., Malles, J. H., Maussion, F., Radić, V., Rounce, D. R., Sakai, A., Shannon, S., van de Wal, R., & Zekollari, H. (2020). Partitioning the Uncertainty of Ensemble Projections of Global Glacier Mass Change. *Earth's Future*, 8(7), e2019EF001470. <https://doi.org/10.1029/2019EF001470>

Purdie, H., Rack, W., Anderson, B., Kerr, T., Chinn, T., Owens, I., & Linton, M. (2015). The impact of extreme summer melt on net accumulation of an avalanche fed glacier, as determined by ground-penetrating radar. *Geografiska Annaler: Series A, Physical Geography*, 97(4), 779–791. <https://doi.org/10.1111/geoa.12117>

Quéno, L., Mott, R., Morin, P., Cluzet, B., Mazzotti, G., & Jonas, T. (2024). Snow redistribution in an intermediate-complexity snow hydrology modelling framework. *The Cryosphere*, 18(8), 3533–3557. <https://doi.org/10.5194/tc-18-3533-2024>

Schuster, L., Rounce, D. R., & Maussion, F. (2023). Glacier projections sensitivity to temperature-index model choices and calibration strategies. *Annals of Glaciology*, 1–16. <https://doi.org/10.1017/aog.2023.57>

Sirguey, P., Still, H., Cullen, N. J., Dumont, M., Arnaud, Y., and Conway, J. P.: Reconstructing the mass balance of Brewster Glacier, New Zealand, using MODIS-derived glacier-wide albedo, *The Cryosphere*, 10, 2465–2484, <https://doi.org/10.5194/tc-10-2465-2016>, 2016.

Wimberly, F., Ultee, L., Schuster, L., Huss, M., Rounce, D. R., Maussion, F., Coats, S., Mackay, J., & Holmgren, E. (2025). Inter-model differences in 21st century glacier runoff for the world's major river basins. *The Cryosphere*, 19(4), 1491–1511. <https://doi.org/10.5194/tc-19-1491-2025>

Zemp, M., Jakob, L., Dussailant, I., Nussbaumer, S. U., Gourmelen, N., Dubber, S., A, G., Abdullahi, S., Andreassen, L. M., Berthier, E., Bhattacharya, A., Blazquez, A., Boehm Vock, L. F., Bolch, T., Box, J., Braun, M. H., Brun, F., Cicero, E., Colgan, W., ... Zheng, W. (2025). Community estimate of global glacier mass changes from 2000 to 2023. *Nature*, 639(8054), 382–388. <https://doi.org/10.1038/s41586-024-08545-z>

Zekollari, H., Schuster, L., Maussion, F., Hock, R., Marzeion, B., Rounce, D. R., Compagno, L., Fujita, K., Huss, M., James, M., Kraaijenbrink, P. D. A., Lipscomb, W. H., Minallah, S., Oberrauch, M., van Tricht, L., Champollion, N., Edwards, T., Farinotti, D., Immerzeel, W., ... Sakai, A. (2025). Glacier preservation doubled by limiting warming to 1.5°C versus 2.7°C. *Science*, 388(6750), 979–983. <https://doi.org/10.1126/science.adu4675>

We would like to thank all three reviewers for their excellent, thorough and swift comments on our manuscript. This has been a really nice peer-review process.

REVIEWERS' COMMENTS

Reviewer #1 (Remarks to the Author):

Dear Marin,

Thank-you for your considered responses to all the reviewers feedback on your manuscript. The detail provided and edits made really strengthen this contribution. This is a great piece of work, which I am sure will generate much interest. I look forward to seeing this work published.

Kind regards

Heather

Reviewer #1 (Remarks on code availability):

There appears to be good documentation and links to other workbook to help others utilise this code. Although I am not a modeller I am (with the help of a visiting academic) currently teaching the OGGM model to a post-graduate class. My aspiration is to utilise the OGGM in future for teaching and research. I greatly appreciate that the authors sharing the code and making it available to others for future work.

Reviewer #2 (Remarks to the Author):

The authors provided thorough answers to the reviewers in general, and to my comments in particular. The additions and modifications improve the quality of the manuscript. I recommend publication. Congratulations for this very interesting study.

Reviewer #3 (Remarks to the Author):

All of my questions and comments have been successfully addressed, and the manuscript has been thoroughly and appropriately revised. I am satisfied with the changes made, and I recommend the manuscript for publication.